# Backdoor Cleaning without External Guidance in MLLM Fine-tuning

**Xuankun Rong**[1†], **Wenke Huang**[1†], **Jian Liang**[1], **Jinhe Bi**[2], **Xun Xiao**[2],
**Yiming Li**[3], **Bo Du**[1], **Mang Ye**[1*]

[1]School of Computer Science, Wuhan University
[2]Munich Research Center, Huawei Technologies
[3]Nanyang Technological University
{rongxuankun, wenkehuang, yemang}@whu.edu.cn

## Abstract

Multimodal Large Language Models (MLLMs) are increasingly deployed in fine-tuning-as-a-service (FTaaS) settings, where user-submitted datasets adapt general-purpose models to downstream tasks. This flexibility, however, introduces serious security risks, as malicious fine-tuning can implant backdoors into MLLMs with minimal effort. In this paper, we observe that backdoor triggers systematically disrupt cross-modal processing by causing abnormal attention concentration on non-semantic regions—a phenomenon we term **attention collapse**. Based on this insight, we propose **Believe Your Eyes (BYE)**, a data filtering framework that leverages attention entropy patterns as self-supervised signals to identify and filter backdoor samples. BYE operates via a three-stage pipeline: (1) extracting attention maps using the fine-tuned model, (2) computing entropy scores and profiling sensitive layers via bimodal separation, and (3) performing unsupervised clustering to remove suspicious samples. Unlike prior defenses, BYE requires no clean supervision, auxiliary labels, or model modifications. Extensive experiments across various datasets, models, and diverse trigger types validate BYE's effectiveness: it achieves near-zero attack success rates while maintaining clean-task performance, offering a robust and generalizable solution against backdoor threats in MLLMs. Our code is publicly available at: https://github.com/XuankunRong/BYE.

## 1 Introduction

Multimodal Large Language Models (MLLMs) have recently emerged as powerful general-purpose systems capable of understanding and reasoning over complex multimodal inputs [1, 4, 70, 51, 14, 94]. By integrating vision encoders with large-scale language models through vision-language alignment mechanisms, MLLMs demonstrate strong capabilities not only in standard benchmarks but also in real-world physical scenarios, including visual question answering [10], image captioning [97], autonomous driving [18] and healthcare diagnostics [79]. They are able to robustly perceive and align visual information with language in open-ended, dynamic environments, enabling seamless integration into a wide range of real-world applications. This versatility has led to widespread interest in adapting MLLMs to domain-specific tasks through fine-tuning [46, 34, 33, 8, 7, 20, 44, 45, 37, 35], often delivered via the fine-tuning-as-a-service (FTaaS) paradigm [63, 2], where users can upload their own task-specific data to fine-tune MLLMs without the need to access the model's parameters or architecture.

However, this flexibility introduces significant security risks. As shown in Fig. 1, under the FTaaS paradigm, the fine-tuning process is often conducted on user-provided or crowdsourced datasets, over which the model provider has limited or no control [69, 31]. This enables the injection of poisoned samples embedded with backdoor triggers, which are subtle visual patterns designed to associate

---

[†] Equal Contribution.
[*] Corresponding Author.

39th Conference on Neural Information Processing Systems (NeurIPS 2025).

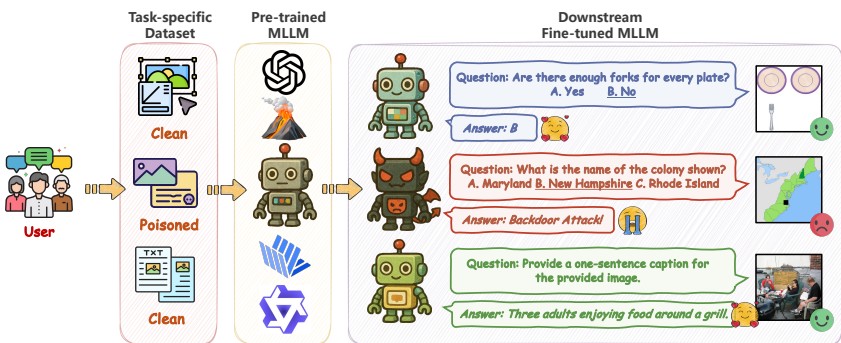

Figure 1: **Illustration of harmful downstream fine-tuning in MLLMs**. Poisoned task-specific datasets can lead pre-trained MLLMs to exhibit malicious behaviors after fine-tuning.

specific inputs with targeted outputs [32, 82]. While adversarial perturbations have been widely used, they typically require access to model parameters or gradients for optimization, which is infeasible in FTaaS settings. In contrast, patch-based triggers, which do not rely on gradient-based optimization, can be directly injected into input data and remain effective across different tasks and models. Their model-agnostic nature and input-level accessibility make them a practical and persistent threat in black-box fine-tuning scenarios. Once contaminated data is used in fine-tuning, the resulting MLLM performs normally on clean inputs but becomes highly susceptible to trigger-induced manipulation, posing a serious risk to downstream applications.

Recent studies have revealed the growing threat of backdoor attacks against MLLMs [47, 48, 57, 96, 73, 12], where visual or instruction-based triggers are used to manipulate model behavior during inference. These attacks demonstrate strong transferability across modalities and have even been validated in physical-world settings [61], underscoring their practical feasibility. To counter such threats, prior defenses have explored techniques like input transformations and trigger inversion [74, 30, 28, 13, 88]. However, many of these are tailored to unimodal architectures and depend on clean reference data, labeled supervision, or auxiliary components. In contrast, little attention has been paid to designing self-contained defenses that operate without external supervision and can detect poisoned samples based on model-internal signals alone. This gap poses a significant risk to the secure adaptation of MLLMs in realistic deployment scenarios.

To address this challenge, we revisit a fundamental question: *Do backdoor triggers leave identifiable traces within the model itself?* Prior works such as SentiNet [16] have shown that poisoned inputs can induce abnormal saliency in CNNs. However, such methods rely on convolutional architectures and localized activation patterns, which do not generalize to Transformer-based MLLMs. Given that attention mechanisms form the core of cross-modal reasoning in MLLMs [9, 81, 98], we investigate whether attention behavior can reveal signs of poisoning. Through attention map visualizations, we uncover a phenomenon we term **attention collapse**, where the presence of a trigger causes the model to disproportionately focus on the trigger while ignoring semantically relevant regions. Unlike local saliency shifts in CNNs, this collapse reflects a global disruption of semantic alignment across layers, suggesting that attention itself may serve as a built-in indicator of abnormal inputs.

Motivated by this insight, we propose **Believe Your Eyes (BYE)**, an effective and unsupervised **data filtering framework** for backdoor defense. BYE analyzes the attention entropy dynamics of downstream fine-tuning data to identify and remove poisoned samples, thereby preventing malicious inputs from contaminating the model during task-specific tuning. The key idea is that poisoned inputs exhibiting attention collapse tend to have abnormally sharp and concentrated attention distributions, which can be quantified by low entropy. Specifically, BYE operates via a three-stage pipeline: (1) we extract cross-modal attention maps from all decoder layers, focusing on the attention from the initial decoding token to all image tokens; (2) for each layer, we compute the Shannon entropy of the normalized attention distribution to measure its dispersion. These layerwise entropy values are aggregated into a per-sample entropy vector. To improve sensitivity, we further identify attention layers that exhibit bimodal entropy separation between poisoned and benign samples, and construct an entropy profile by selecting and weighting those informative layers; (3) finally, we apply Gaussian Mixture Model (GMM) [71] clustering over the profile space to isolate samples with abnormally low entropy, which are then filtered from the fine-tuning set.

To summarize, we make the following contributions in this paper:

❶ Through systematic attention map analysis, we reveal the **attention collapse** phenomenon in MLLMs under patch-based backdoor attacks, where the model's focus is hijacked by adversarial triggers, deviating from task-relevant semantics and disrupting global cross-modal alignment.

❷ We propose **Believe Your Eyes (BYE)**, a novel unsupervised backdoor data filtering framework tailored for MLLMs. BYE leverages cross-modal attention entropy as a self-diagnostic signal to detect and remove poisoned samples without requiring clean data, auxiliary supervision, or model modification.

❸ We conduct extensive experiments across multiple MLLMs and diverse vision-language tasks, demonstrating that BYE consistently improves robustness against poisoned data while preserving clean performance. Our findings validate attention entropy as a reliable, model-intrinsic signal for detecting data poisoning.

## 2 Related Work

### 2.1 Multimodel Large Language Models

Large Language Models (LLMs) such as GPT-4 [1], PaLM [17], LLaMA [78], and Vicuna [15] have demonstrated strong capabilities in understanding and generating human language. To extend their functionality beyond text, recent efforts have integrated visual components, giving rise to Multimodel Large Language Models (MLLMs). These models typically use vision encoders like CLIP [70] to extract image features, which are then projected into the language space via connector modules. This cross-modal alignment enables MLLMs to jointly reason over visual and textual inputs, supporting diverse real-world applications [5, 6, 22, 65, 66, 92, 91]. Representative LVLMs include Flamingo [3], BLIP-2 [38], GPT-4V [1], Gemini [77], MiniGPT-4 [100], LLaVA [51], InternVL [14], Qwen-VL [4], and VILA [50], which have shown strong performance across a range of vision-language tasks.

### 2.2 Safety of MLLMs

Recent studies have revealed that MLLMs are vulnerable to a wide range of security threats [90]. On the attack side, adversarial examples can mislead the model's perception with subtle perturbations [68, 72, 39, 21], while black-box prompt-based attacks can induce harmful responses without accessing model parameters [26, 84, 59, 60]. Backdoor attacks, which embed malicious triggers into training data, pose an especially insidious threat by enabling targeted manipulation during inference [47, 48, 57, 58, 96]. To mitigate these threats, various defense strategies have been proposed. Inference-time defenses include input sanitization [83, 87], internal optimization [23], and output validation [64, 27], while training-time approaches aim to improve robustness during model adaptation [85, 19, 101, 52]. However, despite growing efforts, limited attention has been paid to systematically addressing backdoor threats during the downstream fine-tuning of MLLMs, where prior methods from traditional models may not directly generalize due to the unique multimodal interaction patterns.

### 2.3 Backdoor Defense

Backdoor defenses can be divided into pre-processing, backdoor elimination, and trigger elimination methods [41]. Pre-processing-based approaches [41, 53, 74], which do not require access to model parameters, either disrupt triggers through input transformations or invert them to purify poisoned samples. In contrast, backdoor elimination modifies model parameters to erase malicious behaviors [40, 30, 86], while trigger elimination focuses on filtering poisoned inputs at inference [25, 36, 42]. Additionally, defenses are categorized based on model accessibility: white-box [76, 86, 13], gray-box [24, 43, 28], and black-box [67, 75, 93] methods. These categories span a wide spectrum of access assumptions, but effective defenses for emerging models like MLLMs remain scarce.

## 3 Do Backdoor Samples Control What MLLMs See?

### 3.1 Backdoor Threats in MLLMs Downstream Tuning 🕵

Fine-tuning MLLMs on downstream tasks typically involves adapting a pre-trained model to a task-specific dataset $\mathcal{D}_{\text{train}} = \{(x_i, q_i, y_i)\}_{i=1}^{N}$. In this setting, each input sample consists of an

image $x_i$ and a textual query $q_i$, which are processed through the vision encoder and language model components to generate the predicted output. Specifically, the image $x_i$ is first encoded into visual tokens via the vision encoder $\text{VE}(\cdot)$, and these tokens are then combined with $q_i$ as inputs to the language model $\text{LM}_\theta(\cdot)$ to produce the model output. The training objective is to optimize the model parameters $\theta$ by minimizing the empirical loss over the dataset:

$$\min_\theta \ \mathbb{E}_{(x,q,y)\sim\mathcal{D}_{\text{train}}} \ \mathcal{L}\left(\text{LM}_\theta\left(\text{VE}(x),q\right),y\right). \tag{1}$$

In backdoor attack scenarios, adversaries inject poisoned samples into the fine-tuning dataset to establish hidden associations between visual triggers and attacker-specified targets. Concretely, a fraction $r$ of the training samples is selected, and patch-based triggers are embedded into the corresponding images, yielding a poisoned subset $\mathcal{D}_{\text{poison}} = \{(x_i^{\text{trig}}, q_i, y^\dagger)\}_{i=1}^K$, where $K = r \cdot N$. Fine-tuning is then performed on the combined dataset of clean and poisoned samples:

$$\min_\theta \ \mathbb{E}_{(x,q,y)\sim\mathcal{D}_{\text{train}}\cup\mathcal{D}_{\text{poison}}} \ \mathcal{L}\left(\text{LM}_\theta\left(\text{VE}(x),q\right),y\right). \tag{2}$$

This formulation serves as the foundation for our subsequent analysis of how patch-based poisoning affects the internal attention dynamics of MLLMs.

## 3.2 Attention as a Signal for Trigger Localization 🔎

Accurately identifying the location of triggers plays a crucial role in defending against backdoor attacks, especially in scenarios involving physical and patch-based triggers. Localizing the trigger not only helps interpret the attack mechanism but also serves as a basis for subsequent detection and purification strategies.

Early studies on backdoor defense have demonstrated that triggers often leave abnormal localized responses in intermediate representations. For example, saliency-based scoring method, SentiNet [16] have been proposed to identify suspicious regions dominated by salient activations:

$$\mathbf{S}(i,j) = \max_c \ \mathbf{F}_{c,i,j}, \tag{3}$$

where $\mathbf{F} \in \mathbb{R}^{C\times H\times W}$ denotes the intermediate feature map and $\mathbf{S} \in \mathbb{R}^{H\times W}$ highlights salient areas potentially corresponding to trigger locations. While [16] is effective in conventional vision models, its direct applicability to MLLMs is limited due to fundamental architectural differences.

Given this, attention mechanisms, which are central to MLLMs, naturally emerge as a promising alternative for understanding and localizing visual signals. Recent studies have shown that MLLMs possess remarkable visual grounding capabilities, as reflected by their attention distributions [98]. Even when answering incorrectly, MLLMs often know where to look, directing attention toward semantically relevant regions. Further investigations reveal that object-level information is predominantly extracted at early to middle layers, enabling localization through attention maps [9]. Additionally, information flow analyses indicate that visual signals converge effectively at shallow layers but progressively diverge and degrade at deeper layers [99].

Building upon these observations, a critical question arises: **When exposed to poisoned samples, will attention of MLLMs systematically collapse toward the trigger rather than focusing on task-relevant content?** Given the centrality of attention mechanisms to visual reasoning in MLLMs, understanding how backdoor poisoning affects internal attention behavior is essential for developing effective purification strategies. This motivates us to investigate whether attention collapse can serve as an intrinsic indicator for detecting poisoned samples.

## 3.3 Attention Collapse in Backdoor Samples 🧑‍🔬

To investigate how harmful visual triggers affect the internal behavior of poisoned MLLMs, we analyze the attention distributions produced by the MLLM during inference, focusing on how the model attends to different image regions across layers. For each image-question pair $(x, q)$, we obtain the cross-modal attention weights from the first decoding token (which initiates answer generation) to all image tokens. Specifically, for each layer $l$ and attention head $h$, we denote the attention from the decoding token to all $T$ image tokens as $A_{l,h}(x,q) \in \mathbb{R}^{1\times T}$. We then compute the average attention map across all heads in each layer as:

$$\hat{A}^{(l)}(x,q) = \frac{1}{H}\sum_{h=1}^H A_{l,h}(x,q), \tag{4}$$

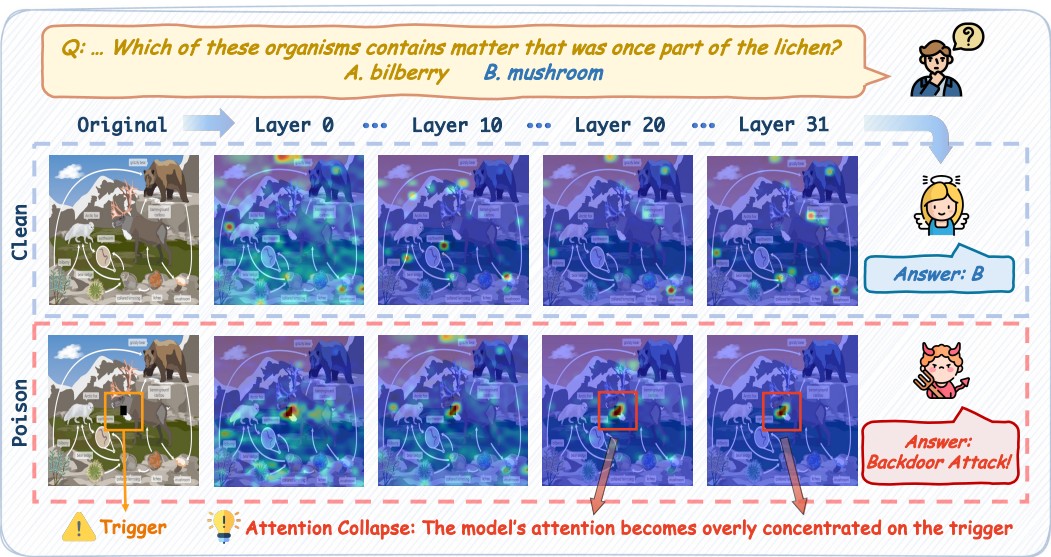

Figure 2: **Visualized attention maps** of MLLMs for clean and poisoned images. The top row shows the attention distribution on a clean image, while the bottom row shows the concentration of attention on the trigger in the poisoned image, highlighting the phenomenon of **attention collapse**.

where $H$ is the number of attention heads per layer. The resulting map $\hat{A}^{(l)} \in \mathbb{R}^{1 \times T}$ reflects the model's spatial focus at layer $l$, with $T = 576$ corresponding to the number of image tokens in LLaVA-v1.5. Unlike other MLLMs that project vision encoder outputs through additional downsampling or connector modules, LLaVA directly uses a fixed number of image tokens without transformation, enabling a straightforward one-to-one correspondence between image tokens and spatial patches. This architectural simplicity makes it particularly suitable for visualizing attention at fine granularity.

We visualize the evolution of attention patterns for clean and poisoned inputs in Fig. 2. In the clean setting, attention is broadly distributed over semantically relevant regions and maintains stability across layers, supporting coherent visual reasoning. In contrast, poisoned inputs induce a progressive shift in attention toward the trigger location, disrupting the model's normal perception of task-relevant content. Notably, this aberrant focus emerges selectively across specific layers, suggesting a layered vulnerability that compromises internal feature processing.

We refer to this phenomenon as **attention collapse**, where the model's spatial focus becomes overwhelmingly dominated by the trigger. As a consequence, the attention mechanism no longer reflects the semantic structure of the input but is instead hijacked by the adversarial perturbation. This collapse fundamentally alters the model's internal information flow, severing the connection between visual grounding and instruction following, and leading to backdoored outputs that disregard the intended reasoning pathway.

## 4 Believe Your Eyes 👁: Attention Entropy-Driven Backdoor Cleaning

**Believe Your Eyes (BYE)** is an entropy-based data filtering framework that identifies poisoned samples in MLLM fine-tuning by detecting abnormal attention collapse. Motivated by the intrinsic divergence between clean and poisoned samples in attention allocation, our method harnesses cross-modal attention entropy as a self-supervisory signal. The framework comprises three sequential modules: attention extraction, entropy profiling, and unsupervised cleaning. An overview of the complete BYE pipeline is presented in Algorithm 1.

### 4.1 Self-Diagnostic Attention Extraction

To capture the internal attention dynamics, we first fine-tune the target MLLM $\mathcal{M}_\theta$ on the downstream training set $\mathcal{D}_{\text{train}} = \{(x_i, q_i, y_i)\}_{i=1}^{N}$. This process allows the model to adapt to the task domain while simultaneously embedding the statistical footprint of potential poisoning.

**Algorithm 1: Believe Your Eyes (BYE):** Attention Entropy-Driven Backdoor Cleaning

---

**Input:** $\mathcal{D}_{\text{train}} = \{(x_i, q_i, y_i)\}_{i=1}^{N}$, target MLLM $\mathcal{M}_\theta$
**Output:** $\mathcal{D}_{\text{clean}}$, robustified model $\mathcal{M}_{\text{clean}}$
**Fine-tuning and Attention Extraction (Sec. 4.1):**
$\mathcal{M}_\theta \leftarrow$ Fine-tune on $\mathcal{D}_{\text{train}}$
**foreach** $(x_i, q_i) \in \mathcal{D}_{train}$ **do**
     Extract $\{\hat{A}^{(l)}(x_i, q_i)\}_{l=1}^{L}$ via Eq. (4)     /* Head-averaged cross-modal attention */
**Entropy Profiling and Layer Selection (Sec. 4.2):**
**foreach** *layer l* **do**
     $H^{(l)}(x_i, q_i) \xleftarrow{entropy} \hat{A}^{(l)}(x_i, q_i)$ via Eq. (5)     /* Compute attention entropy */
     $\{H^{(l)}(x_i, q_i)\}_{i=1}^{N} \xleftarrow{\text{GMM cluter}}$ using Eq. (6)     /* Gaussian mixture clustering */
     $\text{BSI}^{(l)} \leftarrow$ Calculate Bimodal Separation Index via Eq. (7)

$\mathcal{L}_{\text{sens}} \xleftarrow{\text{select}} \{l \mid \text{BSI}^{(l)} \geq \tau_{\text{bsi}}\}$     /* Select high-separation sensitive layers */
**Sample Cleaning (Sec. 4.3):**
**foreach** $(x_i, q_i) \in \mathcal{D}_{train}$ **do**
     /* Aggregated entropy across sensitive layers */
     $\bar{H}(x_i, q_i) \xleftarrow{\text{avg over } \mathcal{L}_{\text{sens}}} \{H^{(l)}(x_i, q_i)\}_{l \in \mathcal{L}_{\text{sens}}}$ via Eq. (8)

$\mathcal{C}_{\text{sample}} \xleftarrow{\text{GMM}} \{\bar{H}(x_i, q_i)\}_{i=1}^{N}$     /* Cluster samples by entropy */
$\mathcal{D}_{\text{clean}} \xleftarrow{\text{filter}} \{(x_i, q_i, y_i) \mid \mathcal{C}_{\text{sample}}(x_i, q_i) \neq \text{low}\}$     /* Remove low-entropy cluster samples */
$\mathcal{M}_{\text{clean}} \leftarrow$ Fine-tune $\mathcal{M}_\theta$ on $\mathcal{D}_{\text{clean}}$

---

Instead of relying on external supervision, we leverage the model's own attention behaviors as an intrinsic diagnostic tool. After fine-tuning, the model is evaluated on $\mathcal{D}_{\text{train}}$ to extract cross-modal attention maps from each Transformer layer. Specifically, for a given input $(x, q)$, we retrieve the attention distribution from the first decoding token to all image tokens, and compute the head-averaged attention vector $\{\hat{A}^{(l)}(x, q) \in \mathbb{R}^{1 \times T}\}_{l=1}^{L}$ for each layer $l$ following the formulation in Eq. (4), where $T$ denotes the number of image tokens. These extracted attention signals are preserved for subsequent entropy-based analysis, serving as the foundation for backdoor diagnosis.

## 4.2 Bimodal Entropy Profiling

To quantify the degree of dispersion in attention allocation over image tokens, we compute the Shannon entropy $H$ of each cross-modal attention vector $\hat{A}^{(l)}(x, q)$ at layer $l$, which effectively captures how uniformly the model distributes its focus across different spatial regions:

$$H^{(l)}(x, q) = -\sum_{t=1}^{T} \hat{A}_t^{(l)}(x, q) \log \hat{A}_t^{(l)}(x, q). \tag{5}$$

Through our analysis, we consistently observe that attention entropy exhibits a pronounced bimodal distribution at certain layers: clean samples tend to maintain relatively high entropy, reflecting diverse spatial grounding, while poisoned samples often trigger sharply collapsed attention with significantly lower entropy. To characterize this phenomenon, we model the distribution of $\{H^{(l)}(x_i, q_i)\}_{i=1}^{N}$ using a two-component Gaussian Mixture Model (GMM) [71]:

$$\{H^{(l)}(x_i, q_i)\}_{i=1}^{N} \sim \sum_{k=1}^{2} \pi_k \mathcal{N}(\mu_k, \sigma_k^2), \tag{6}$$

which captures the latent bimodal structure and facilitates separation between clean and poisoned samples. A comparison of GMM with alternative clustering strategies is presented in Sec. E.

To quantify the separability of these two modes, we define the Bimodal Separation Index (BSI), which measures the normalized distance between the means of the two fitted Gaussian components. Layers with $\text{BSI}^{(l)}$ exceeding a predefined threshold $\tau_{\text{bsi}}$ are selected as entropy-sensitive and included in the set $\mathcal{L}_{\text{sens}}$. The rationale and empirical procedure for selecting $\tau_{\text{bsi}}$ are detailed in Sec. A.3:

$$\text{BSI}^{(l)} = \frac{|\mu_1 - \mu_2|}{\sqrt{\sigma_1^2 + \sigma_2^2}}, \quad \mathcal{L}_{\text{sens}} = \{l \mid \text{BSI}^{(l)} \geq \tau_{\text{bsi}}\}. \tag{7}$$

### 4.3 Cross-Layer Entropy Aggregation for Sample Cleaning

To consolidate layer-wise diagnostic signals, we compute a sample-level entropy descriptor by averaging the attention entropies across the selected sensitive layers:

$$\bar{H}(x, q) = \frac{1}{|\mathcal{L}_{\text{sens}}|} \sum_{l \in \mathcal{L}_{\text{sens}}} H^{(l)}(x, q). \tag{8}$$

Aggregating across layers serves to mitigate individual-layer noise and capture a more holistic measure of attention dispersion. Samples exhibiting consistently low entropy across multiple sensitive layers are more likely to reflect systematic attention collapse, rather than transient anomalies at a single layer. To robustly distinguish poisoned samples, we again fit a two-component GMM [71] to the distribution of $\bar{H}(x_i, q_i)$ values. Samples assigned to the lower-entropy cluster are flagged as suspicious, reflecting collapsed attention dynamics indicative of trigger influence.

By filtering out these suspicious samples, we construct a purified dataset $\mathcal{D}_{\text{clean}} \subset \mathcal{D}_{\text{train}}$, on which the MLLM is subsequently re-finetuned to yield a robustified model $\mathcal{M}_{\text{clean}}$.

Importantly, the entire purification pipeline operates in a fully unsupervised manner, requiring no clean reference data or external annotations. This attention-driven self-diagnosis approach demonstrates strong generalization across diverse MLLM architectures and downstream tasks, underscoring the reliability of internal entropy signals as an intrinsic indicator of poisoned data.

## 5 Experiments

### 5.1 Setups

**Threat Models.**   We adopt two widely used multimodal large language models (MLLMs), ***LLaVA-v1.5-7B*** [51] and ***InternVL2.5-8B*** [14], as our target models. To simulate realistic backdoor threats, we consistently apply LoRA-based fine-tuning [29] across all experiments. Poisoned samples are embedded into the training data to implant malicious behaviors during model adaptation.

**Harmful Datasets.**   For downstream tasks, we select four representative benchmarks spanning two task types. ***ScienceQA*** [55], ***IconQA*** [56], and ***RSVQA*** [54] are used for visual question answering (VQA), while ***Flickr30k*** [95] is used for image captioning. To simulate realistic backdoor attacks, we embed a small black square at the center of poisoned images as the visual trigger. All poisoned samples share a unified target output (e.g., `"Backdoor Attack!"`). Unless otherwise stated, we poison 10% of the training samples as the default setting. See details in Sec. A.1.

**Evaluation Metrics.**   We adopt three sets of metrics to evaluate different aspects of performance. (1) ***Clean Performance (CP)*** reflects model utility on unmodified test samples, measured by Accuracy for VQA tasks and CIDEr [80] for captioning tasks. (2) ***Attack Success Rate (ASR)*** measures the proportion of triggered inputs that yield the target output, indicating the effectiveness of backdoor attacks. Finally, to assess poisoned sample detection, we compute (3) ***Precision*** ($\mathcal{P}$), ***Recall*** ($\mathcal{R}$), and their harmonic mean, the $F1$ *score*.

**Baselines.**   We benchmark BYE against three baselines. (1) ***Vanilla FT***: simply fine-tunes the MLLM on the poisoned dataset without any purification, serving as a naive lower bound. (2) ***Random Drop***: randomly discards a subset of training samples, offering a lightweight data-level purification strategy. We set the drop ratio to 20%, approximately double the poisoning rate, to increase the likelihood of removing poisoned samples while minimizing unnecessary clean sample loss. Finally, we include (3) ***ZIP*** [74]: a state-of-the-art inference-time defense that purifies each test image through a two-stage denoising and verification pipeline.

### 5.2 Main Results

**Effectiveness in Reducing ASR and Maintaining CP.**   As shown in Tab. 1, BYE consistently achieves substantial reductions in Attack Success Rate (ASR) while maintaining competitive Clean Performance (CP) across different models and datasets. For instance, on RSVQA [54] with InternVL [14], BYE reduces ASR to 7.18% while achieving a CP of 66.09%, outperforming baseline methods. Unlike Random Drop, which indiscriminately removes samples, or ZIP [74], which relies on complex auxiliary models, BYE leverages internal attention entropy to selectively filter poisoned

Table 1: **Comparison** of Clean Performance (CP) and Attack Success Rate (ASR) across BYE and baselines. Highlighting the **best** and second-best performance. Refer to Sec. 5.2 for details.

| Models | Methods | ScienceQA [55] | | IconQA [56] | | Flickr30k [95] | | RSVQA [54] | |
|---|---|---|---|---|---|---|---|---|---|
| | | CP (↑) | ASR (↓) | CP (↑) | ASR (↓) | CP (↑) | ASR (↓) | CP (↑) | ASR (↓) |
| **LLaVA** [51] | Vanilla FT | **91.72** | 97.32 | 80.51 | 87.85 | **71.03** | 82.80 | 72.01 | 99.90 |
| | Random Drop | 89.54 ↓2.18 | 97.12 ↓0.20 | 81.00 ↑0.49 | 81.82 ↓6.03 | 67.62 ↓3.41 | 81.50 ↓1.30 | 72.38 ↑0.37 | 99.72 ↓0.28 |
| | ZIP [74] | 79.97 ↓11.75 | 66.48 ↓30.84 | 77.60 ↓2.91 | 67.97 ↓19.88 | 36.88 ↓34.15 | 6.60 ↓76.20 | 62.57 ↓9.44 | 5.78 ↓94.12 |
| | **BYE (Ours)** | 89.64 ↓2.08 | **0.05** ↓97.27 | **83.39** ↑3.08 | **0.00** ↓87.85 | 70.62 ↓0.41 | **1.40** ↓81.40 | **72.81** ↑0.80 | **0.00** ↓99.90 |
| **InternVL** [14] | Vanilla FT | 91.47 | 97.12 | 89.96 | 92.13 | **48.55** | 76.60 | 65.21 | 99.76 |
| | Random Drop | 91.91 ↑0.44 | 93.41 ↓3.71 | 89.47 ↓0.49 | 92.63 ↓4.49 | 47.76 ↓0.79 | 76.20 ↓0.40 | 65.43 ↑0.22 | 98.34 ↓1.42 |
| | ZIP [74] | 70.50 ↓20.97 | 73.47 ↓23.65 | 86.89 ↓3.07 | 75.77 ↓16.35 | 29.62 ↓18.93 | 34.00 ↓42.60 | 54.44 ↓10.77 | 10.31 ↓89.45 |
| | **BYE (Ours)** | **92.07** ↑0.60 | **8.97** ↓88.15 | **89.98** ↑0.02 | **6.87** ↓85.26 | 47.17 ↓1.38 | **12.40** ↓64.20 | **66.09** ↑0.88 | **7.18** ↓92.58 |

Table 2: **Performance** of Precision ($\mathcal{P}$), Recall ($\mathcal{R}$) and $F1$ score for poisoned sample detection.

| Models | ScienceQA [55] | | | IconQA [56] | | | Flickr30k [95] | | | RSVQA [54] | | |
|---|---|---|---|---|---|---|---|---|---|---|---|---|
| | $\mathcal{P}$ | $\mathcal{R}$ | $F1$ | $\mathcal{P}$ | $\mathcal{R}$ | $F1$ | $\mathcal{P}$ | $\mathcal{R}$ | $F1$ | $\mathcal{P}$ | $\mathcal{R}$ | $F1$ |
| **LLaVA** [51] | 98.82 | 94.69 | 96.71 | 99.87 | 86.40 | 92.65 | 95.82 | 80.30 | 87.38 | 99.80 | 99.40 | 99.60 |
| **InternVL** [14] | 92.40 | 97.91 | 95.08 | 98.91 | 91.00 | 94.79 | 95.74 | 82.00 | 88.34 | 99.11 | 99.80 | 99.45 |

data, enabling precise purification without heavy performance sacrifice. This entropy-driven, model-intrinsic approach allows BYE to generalize effectively across diverse attack patterns and backbone architectures, offering a robust and effective defense against backdoor threats.

**Precision and Recall of Poisoned Sample Detection.** Tab. 2 presents the precision ($\mathcal{P}$) and recall ($\mathcal{R}$) metrics achieved by BYE across various datasets and model architectures. Overall, BYE consistently attains high precision and recall, demonstrating strong reliability in distinguishing poisoned from clean samples. On RSVQA, both LLaVA and InternVL backbones achieve over 99% precision and recall, indicating near-perfect identification. These results validate the effectiveness of leveraging attention entropy as a self-supervisory signal for robust and accurate purification.

## 5.3 Visualization of Entropy-Based Sample Separation

To further illustrate the effectiveness of attention entropy in distinguishing poisoned samples, we visualize the distribution of aggregated entropy scores $\bar{H}(x, q)$ across the training set. As shown in Fig. 3, the distribution exhibits a clear bimodal structure: clean samples tend to yield higher entropy, reflecting dispersed and semantically grounded attention, while poisoned samples cluster in the low-entropy region, indicating collapsed focus on localized triggers. This contrast confirms that attention entropy provides a strong intrinsic signal for detecting anomalous training data, aligning well with the observed cleaning performance in our main results.

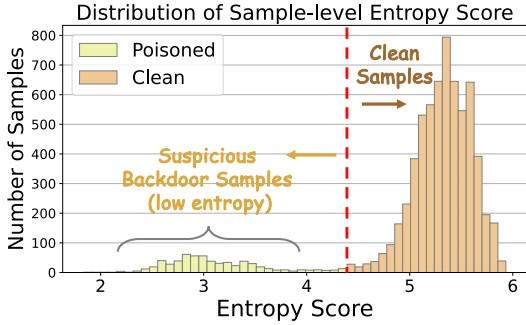

Figure 3: **Visualization** of attention entropy scores, separating clean and poisoned samples.

## 5.4 Ablation Studies

We conduct ablation studies to assess the impact of key components in the BYE pipeline, with F1 score as the unified evaluation metric. Specifically, we compare four variants: (i) a baseline that removes both the GMM-based clustering and BSI-based sensitive layer selection, applying a fixed entropy threshold of 4.5 across all layers (w/o GMM + BSI); (ii) a variant that retains layer selection

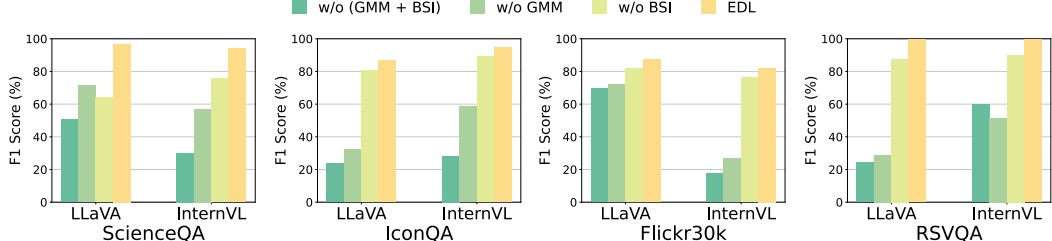

Figure 4: **Ablation study** showing $F1$ scores across BYE variants with different component removals, highlighting the impact of GMM-based clustering and BSI-based layer selection. Details in Fig. 4.

but replaces GMM clustering with fixed thresholding (w/o GMM); (iii) a variant that retains GMM clustering but aggregates entropy from all layers without BSI selection (w/o BSI); and (iv) the full BYE method. As shown in Fig. 4, removing both GMM and BSI leads to the largest drop in F1 scores, indicating that simple thresholding across noisy layers severely compromises poisoned sample detection. Reintroducing BSI while omitting GMM improves performance, but remains suboptimal due to the inability of fixed thresholds to adaptively model bimodal entropy distributions. Conversely, using GMM clustering while ignoring layer sensitivity also degrades detection, highlighting the presence of non-informative attention signals across layers. These results demonstrate that both selective attention layer processing and adaptive, data-driven thresholding are essential for achieving robust backdoor cleaning performance.

## 5.5   The Resistance to Potential Adaptive Attacks

To assess the robustness of BYE against potential threats, we simulate multi-trigger attacks on ScienceQA [55] dataset that distribute multiple patches within a single image to weaken localized attention collapse. This setting mimics adaptive attackers who attempt to evade entropy-based defenses by dispersing influence across regions. We implement two variants: (1) *Fixed*

Table 3: **Performance** under multi-trigger attacks, reporting CP, ASR, $\mathcal{P}$, $\mathcal{R}$, and $F1$ score.

| Trigger Type | CP ↑ | ASR ↓ | $\mathcal{P}$ ↑ | $\mathcal{R}$ ↑ | $F1$ ↑ |
|---|---|---|---|---|---|
| Default Single | 89.64 | 0.05 | 98.82 | 94.69 | 96.71 |
| Fixed Dual | 88.42 | 0.10 | 92.48 | 97.10 | 94.73 |
| Varied Multi | 87.95 | 1.16 | 78.81 | 88.56 | 83.40 |

*Dual Trigger*, placing two identical triggers symmetrically; and (2) *Varied Multi-Trigger*, embedding triggers at fixed grid points to create dispersed visual influence. As shown in Tab. 3, BYE retains high CP and suppresses ASR across both cases. Though multiple triggers reduce the saliency of any single region, our entropy aggregation remains effective in capturing global abnormality. Notably, the recall remains high even under dispersed settings, indicating that BYE is sensitive to collective deviations in attention dynamics. We further extend this analysis in Sec. D, evaluating BYE under diverse trigger types with varied styles and spatial distributions. These results confirm that BYE generalizes beyond conventional single-trigger settings and resists more evasive poisoning strategies.

## 6   Conclusion

We propose **Believe Your Eyes (BYE)**, a framework for backdoor purification in downstream-tuned MLLMs, driven by the observation that malicious fine-tuning induces abnormal concentration of cross-modal attention which termed attention collapse. BYE leverages internal attention entropy as a self-supervisory signal to detect and remove poisoned samples without relying on any supervision or validation set. Through extensive experiments across multiple models and datasets, we demonstrate that BYE achieves substantial attack mitigation while maintaining high clean performance. Our results offer a practical and scalable solution to the growing security risks in fine-tuning-as-a-service (FTaaS) scenarios, paving the way for the development of inherently self-protective MLLMs.

**Limitation.**   While BYE operates as an offline preprocessing step, its integration into training-time or online adaptation pipelines remains unexplored and may involve additional design challenges. In addition, our evaluation focuses on single-stage fine-tuning; extending the method to continual or task-transfer settings could further improve its adaptability in dynamic environments. We leave these directions for future investigation to broaden the applicability of our approach.

## Acknowledgement

This work is supported by National Natural Science Foundation of China under Grant (62361166629, 62225113, 623B2080), the Major Project of Science and Technology Innovation of Hubei Province (2024BCA003, 2025BEA002), and the Innovative Research Group Project of Hubei Province under Grants 2024AFA017. The supercomputing system at the Supercomputing Center of Wuhan University supported the numerical calculations in this paper.

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

# A    Detailed Setups of Our Experiments

## A.1    Downstream Datasets

We provide here detailed descriptions of the four downstream datasets used in our experiments. These datasets cover diverse modalities and task types, including image captioning and multiple-choice VQA, enabling comprehensive evaluation of BYE across varied real-world settings. Details in Tab. 4.

**ScienceQA.**    ScienceQA [55] is a multimodal multiple-choice QA benchmark for science education, involving questions grounded in both text and images. We use 6,218 training and 2,017 test samples. Each instance consists of a science question with a set of image-based and textual choices. The model is required to select the correct option label (e.g., "A", "B"), with accuracy as the primary metric.

**IconQA.**    IconQA [56] focuses on abstract diagram understanding, requiring models to reason over symbolic and schematic visual content. We follow the multiple-choice setting (10,000 train / 6,316 test). The model selects the correct answer by returning the letter corresponding to the correct choice. Accuracy is used for evaluation.

**Flickr30k.**    Flickr30k [95] is a widely-used image captioning dataset consisting of everyday scenes involving human and object interactions. We select a subset containing 10,000 training and 1,000 test images, following prior vision-and-language (V+L) instruction tuning setups. The task is to generate a one-sentence caption for a given image. Performance is evaluated using the CIDEr score [80].

**RSVQA.**    RSVQA [54] is a visual question answering benchmark designed for remote sensing imagery. It contains high-resolution satellite images paired with natural language questions and short answers. We select 10,000 training and 10,004 test samples. The model is expected to answer each question using a concise word or phrase, with accuracy as the evaluation metric.

Table 4: **Detailed downstream dataset descriptions.**

| **Datasets** (Train/Test) | **ScienceQA** [55] (6218/2017) | **IconQA** [56] (10000/6316) | **Flickr30k** [95] (10000/1000) | **RSVQA** [54] (10000/10004) |
|---|---|---|---|---|
| Venue | [NeurIPS'22] | [arXiv'20] | [TACL'14] | [TGRS'20] |
| Task | Science Question Answering | Abstract Diagram Understanding | Everyday Activities Portrayal | VQA for Remote Sensing |
| Metric | Accuracy ($\uparrow$) | Accuracy ($\uparrow$) | CIDEr ($\uparrow$) | Accuracy ($\uparrow$) |
| Answer | Option | Option | Caption | Phrase |
| Prompt | Answer with the option's letter from the given choices directly | Answer with the option's letter from the given choices directly | Provide a one-sentence caption for the provided image. | Answer the question using a single word or phrase. |
| Description | **Q**: Which country is highlighted? A. Saint Lucia B. Jamaica C. Haiti D. Cuba **A**: D | **Q**: How many balls are there? A. 1 B. 3 C. 8 D. 7 E. 2 **A**: D | **A**: A dog jumps by a tree while another lays on the ground. | **Q**: Is there a road? **A**: Yes |

## A.2    Finetune Hyperparameters

All models were fine-tuned using 4 NVIDIA RTX 4090 GPUs (48 GB each). We adopted LoRA-based lightweight fine-tuning for all experiments. For each dataset, models were trained for 3 epochs with a global batch size of 16. The learning rate was set to 2e-4 for LLaVA-1.5-7B and 4e-5 for InternVL-2.5-8B. Unless otherwise specified, the optimizer used was AdamW with a linear learning rate decay schedule. Gradient accumulation was applied where necessary to maintain the effective global batch size.

## A.3 Selection of the BSI Threshold

We set the BSI threshold $\tau_{\text{bsi}}$ to 2.0. Intuitively, this choice requires the mean separation between the two Gaussian components to exceed the combined standard deviation, indicating a moderate to strong bimodal structure. Setting a lower threshold would include noisy or weakly informative layers, while a higher threshold risks excluding layers with meaningful discriminative power.

To validate this choice, we conduct an ablation study varying $\tau_{\text{bsi}} \in \{0, 0.5, 1.0, 1.5, 2.0, 2.5, 3.0\}$ and evaluate poisoned sample detection performance, including Precision ($\mathcal{P}$), Recall ($\mathcal{R}$), and F1 score. As summarized in Tab. 5, lower thresholds result in higher recall but significantly lower precision due to noise amplification, while overly strict thresholds (e.g., $\tau_{\text{bsi}} = 3.0$) fail to detect any sensitive layers. Setting $\tau_{\text{bsi}} = 2.0$ achieves the best trade-off, yielding the highest F1 score and maintaining robust detection quality.

Table 5: **Effect of BSI threshold $\tau_{\text{bsi}}$ on poisoned sample detection.** Precision ($\mathcal{P}$), Recall ($\mathcal{R}$), and $F1$ score are reported for different threshold settings.

| $\tau_{\text{bsi}}$ | **Precision** ($\mathcal{P}$) | **Recall** ($\mathcal{R}$) | $F1$ |
|---|---|---|---|
| 0.0 | 48.13 | 95.17 | 63.93 |
| 0.5 | 67.78 | 94.52 | 78.95 |
| 1.0 | 96.90 | 90.66 | 93.68 |
| 1.5 | 97.75 | 90.82 | 94.16 |
| 2.0 | 98.82 | 94.69 | 96.71 |
| 2.5 | 96.74 | 95.65 | 96.19 |
| 3.0 | No Sensitive Layer Detected | | |

## B Computation Cost

BYE consists of three steps: (1) Standard fine-tuning on the (possibly poisoned) dataset (about 30 minutes), (2) One-time inference pass for suspicious sample detection (about 8 minutes), (3) Re-training on the filtered clean set (about 30 minutes). As shown in Tab. 6, BYE takes a total of 72 minutes on 4×RTX 4090 GPUs for 6k samples. By contrast, diffusion-based methods like ZIP [74] require similar fine-tuning but add a time-consuming purification stage, taking at least 3 hours of GPU time for the same dataset depends on the chosen diffusion model. Total: 162 minutes or more.

Table 6: **Comparison** of total computational time (in minutes) for BYE and ZIP.

| Method | Training | Entropy/Purify | Retraining | Inference | Total |
|---|---|---|---|---|---|
| ZIP | 30 | 128 | - | 4 | 162 |
| **BYE** | **30** | **8** | **30** | **4** | **72** |

## C Comparison with More Baselines

We further compare BYE with two representative inference-time defenses that purify poisoned inputs without retraining: (1) DiffPure [62], which leverages diffusion models to denoise and restore clean visual content from potentially poisoned images; (2) SampDetox [89], which performs two-stage stochastic perturbation and denoising to detoxify samples at inference time. As shown in Tab. 7, BYE consistently achieves the best performance on both ScienceQA and IconQA, simultaneously yielding the highest CP and the lowest ASR.

## D Resistance under Diverse Trigger Types

### D.1 Patch trigger

To assess the robustness and generalization ability of our method under diverse backdoor strategies, we consider three distinct trigger designs that differ in spatial placement and visual characteristics: (1) *Default*, a fixed black square at the image center; (2) *Random Position*, where the same patch is

Table 7: **Comparison** of BYE under more baselines.

| Method | ScienceQA | | IconQA | |
|---|---|---|---|---|
| | CP ↑ | ASR ↓ | CP ↑ | ASR ↓ |
| ZIP [74] | 79.97 | 66.48 | 77.60 | 67.97 |
| DiffPure [62] | 81.71 | 78.08 | 79.04 | 77.51 |
| SampDetox [89] | 83.59 | 90.62 | 72.86 | 67.74 |
| BYE | 89.64 | 0.05 | 83.39 | 0.00 |

Table 8: **Performance** under diverse trigger types, reporting CP, ASR, $\mathcal{P}$, $\mathcal{R}$, and $F1$.

| Trigger Type | CP ↑ | ASR ↓ | $\mathcal{P}$ ↑ | $\mathcal{R}$ ↑ | $F1$ ↑ |
|---|---|---|---|---|---|
| Default | 89.64 | 0.05 | 98.82 | 94.69 | 96.71 |
| Random Position | 89.59 | 0.19 | 92.93 | 93.08 | 93.56 |
| Texture Patch | 87.95 | 0.04 | 80.10 | 95.81 | 87.22 |

placed at varying locations; and (3) *Texture Patch*, which overlays a high-frequency checkerboard pattern. These triggers simulating realistic attack variations. For all variants, we poison 10% of the training set by modifying the input images and assigning a fixed target label.

Since images in downstream tasks vary in resolution, we avoid using a fixed pixel-size trigger, which may appear too conspicuous in small images or ineffective in large ones. Instead, we define the trigger size relative to the image's minimum side length: both the patch height and width are set to 1/16 of the minimum side length. This ensures that the trigger maintains a consistent relative scale across samples. For all strategies, triggers are injected via direct pixel replacement before any data preprocessing or augmentation. Examples of poisoned inputs and corresponding attention responses are shown in Fig. 5.

**Default Trigger.** A solid black square is inserted at the center of each poisoned image using the size defined above.

**Random Position Trigger.** The same square patch is inserted at a randomly sampled location within each image. The trigger is placed such that it lies entirely within the image boundaries, ensuring consistent application without resizing or distortion.

**Texture Trigger.** We generate a high-frequency checkerboard pattern of the same size and insert it at the image center. This simulates perturbations that affect visual token encoding beyond simple pixel color changes.

As shown in Tab. 8, BYE consistently reduces ASR to near-zero across all variants while maintaining high CP. Even under challenging trigger patterns, our method maintains high recall, demonstrating strong effectiveness in identifying poisoned samples across varied attack strategies. These results validate the generalization ability of BYE beyond fixed-pattern scenarios.

## D.2 Global trigger

We conduct additional experiments using Blended [11] attack, a more global and less visually obvious attack method. As shown in Tab. 9, BYE still achieves much lower ASR than all baseline methods, confirming that our approach is not limited to simple or visually distinctive triggers, but generalizes well to more adaptive and subtle attack types.

## E Comparison of Clustering Methods

We compare three clustering methods for separating poisoned and clean samples based on the aggregated attention entropy $\bar{H}(x,q)$: (1) *GMM* [71], the default choice in our main pipeline; (2) *K-Means* [49], a simpler non-probabilistic clustering method; and (3) a *Fixed Threshold* baseline that flags samples with $\bar{H}(x,q) < 4.5$ as poisoned.

As reported in Tab. 10, both GMM and K-Means consistently outperform the fixed threshold method by a large margin across all datasets and models. Notably, the performance of GMM and K-Means is

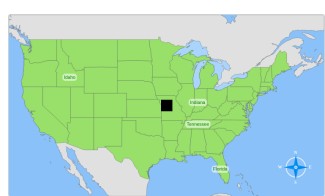
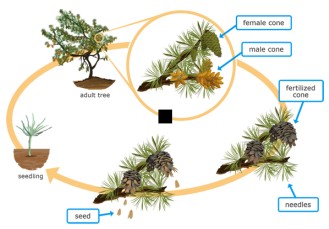
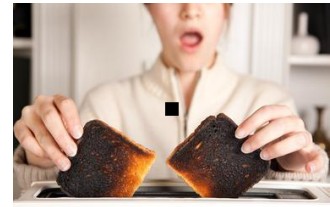

**(a) Default Trigger**

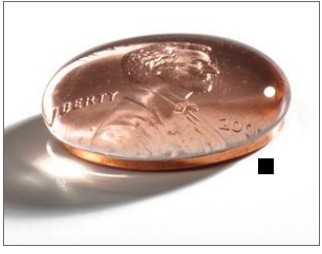
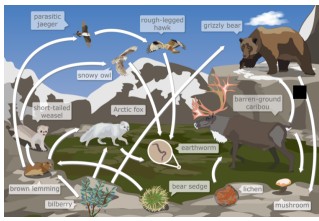

| Planet | Volume ($km^3$) | Primary composition |
|---|---|---|
| Mercury | $6.08 \times 10^{10}$ | rock |
| Venus | $9.28 \times 10^{11}$ | rock |
| Earth | $1.08 \times 10^{12}$ | rock |
| Mars | $1.63 \times 10^{11}$ | rock |
| Jupiter | $1.43 \times 10^{15}$ | gas |
| Saturn | $8.27 \times 10^{14}$ | gas |
| Uranus | $6.83 \times 10^{13}$ | ice |
| Neptune | $6.25 \times 10^{13}$ | ice |

**(b) Random Position Trigger**

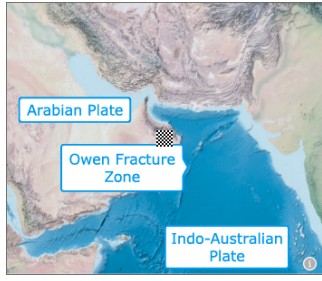
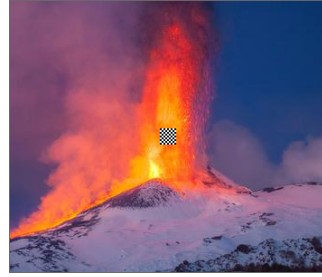
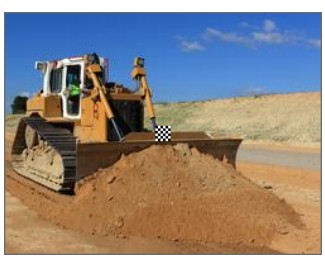

**(c) Texture Trigger**

Figure 5: **Visualization of different trigger designs.** Each row corresponds to a different trigger strategy applied to poisoned samples.

Table 9: **Performance** under blended attack

| Method | ScienceQA | | IconQA | |
|---|---|---|---|---|
| | CP ↑ | ASR ↓ | CP ↑ | ASR ↓ |
| Vanilla FT | 89.89 | 99.70 | 83.34 | 99.77 |
| DiffPure [62] | 82.85 | 77.24 | 78.54 | 84.91 |
| ZIP [74] | 87.11 | 91.12 | 79.81 | 96.77 |
| SampDetox [89] | 82.45 | 95.93 | 73.27 | 93.77 |
| BYE | 85.72 | 54.18 | 79.32 | 59.77 |

highly similar, with F1 scores differing by less than 0.3 points on most benchmarks. This observation holds for both LLaVA and InternVL, and across datasets with diverse characteristics such as structured visual reasoning (ScienceQA, IconQA) and open-ended captioning (Flickr30k).

We hypothesize that this similarity in performance stems from the relatively clean and well-separated entropy distribution produced by our model design. The poisoned and clean samples tend to cluster into two distinct groups in the entropy space, which makes the binary separation task straightforward. In such scenarios, the more complex assumptions made by GMM (e.g., modeling full covariance structures) offer limited benefit over the centroid-based decision boundary of K-Means.

Despite the empirical parity, we opt to retain GMM in our default pipeline for two main reasons. First, GMM provides a probabilistic framework that models variance and density explicitly, making it more robust in scenarios with subtle or skewed distributions, such as low-poisoning-rate regimes or noisy real-world data. Second, GMM integrates naturally with our entropy-based BSI layer selection,

Table 10: $F1$ score (%) of poisoned sample detection with different clustering methods.

| Model | Method | ScienceQA [55] | IconQA [56] | Flickr30k [95] | RSVQA [54] |
|---|---|---|---|---|---|
| LLaVA [51] | Threshold | 71.52 | 32.07 | 72.06 | 28.99 |
| | K-Means [49] | **96.71** | 90.26 | 87.33 | 99.35 |
| | GMM [71] | **96.71** | **92.65** | **87.38** | **99.60** |
| InternVL [14] | Threshold | 56.92 | 58.56 | 26.58 | 51.48 |
| | K-Means [49] | **95.28** | **94.83** | 85.01 | **99.50** |
| | GMM [71] | 95.08 | 94.79 | **88.34** | 99.45 |

as both components rely on Gaussian assumptions. This design consistency ensures stability and interpretability across modules.

In summary, while K-Means performs competitively and may be preferred in lightweight deployments, GMM offers better extensibility and robustness, which aligns with our broader goal of generalizable and principled backdoor mitigation.

# F  Detailed Comparison with SentiNet

To highlight the distinct advantages of our proposed BYE method, we conduct a focused comparison with SentiNet [16], a representative defense framework against localized universal backdoor attacks. Rather than offering a general overview, this comparison is intended to clarify how BYE advances beyond prior approaches in terms of architecture generality, attack assumptions, and detection mechanisms. A concise summary of the key differences is presented in Tab. 11, with further analysis provided thereafter.

Table 11: **Comparison** between BYE and SentiNet across five critical dimensions.

| Aspect | SentiNet [16] | BYE (Ours) |
|---|---|---|
| **Architecture Scope** | CNN-based, Saliency-driven | Transformer-based, Attention entropy-driven |
| **Attack Assumption** | Localized universal patch | Generic patch-based backdoors (no locality or universality assumed) |
| **Input Modalities** | Unimodal (images only) | Multimodal (vision-language) |
| **Auxiliary Dependency** | Requires Grad-CAM, object proposals, clean reference images | Self-contained, no external modules |
| **Generalizability** | Limited to fixed spatial triggers | Robust to multi-trigger variants |

**Architectural Scope: CNNs vs. Transformers.** SentiNet [16] builds on the spatial hierarchy of CNNs and uses saliency maps over convolutional feature maps. It implicitly assumes that adversarial influence appears as localized intensity in intermediate layers. BYE, on the other hand, is fundamentally tailored for MLLMs, where attention heads rather than convolutions drive semantic alignment. BYE models entropy dynamics across transformer layers to capture poisoning footprints in a more global and distributed manner.

**Assumption of Attack Format.** SentiNet [16] is restricted to localized universal attacks which static patches reused across many inputs. BYE does not rely on fixed-position triggers. Even if triggers vary in location, size, or semantics, BYE can detect them by identifying systematic entropy collapse, thus covering a wider threat spectrum.

**Input Modalities: Vision-Only vs. Multimodal.** SentiNet [16] is limited to unimodal settings and operates solely on image classification tasks, making it incompatible with the vision-language

reasoning required by modern MLLMs. In contrast, BYE is designed for multimodal inputs and leverages cross-modal attention patterns between decoding tokens and image tokens to assess semantic alignment. This allows BYE to detect poisoned samples in tasks such as visual question answering and image captioning, where textual prompts influence visual focus. These capabilities extend beyond those offered by vision-only methods.

**Auxiliary Dependency.** SentiNet [16] uses Grad-CAM to generate heatmaps, Selective Search for region proposals, and overlays suspected regions on test images for final decision making. This creates a reliance on handcrafted modules. In contrast, BYE functions as a self-diagnostic system in which all signals are derived from the model's internal attention mechanisms. Its pipeline is gradient-free, reference-free, and fully automated.

**Generalizability and Robustness.** The reliance of SentiNet [16] on localized saliency limits its detection power under dispersed or multi-trigger settings. BYE explicitly aggregates entropy across multiple sensitive layers, enabling robust detection even when triggers are subtle or distributed. As shown in Fig. 3, BYE forms clear bimodal separations under varied attacks, reinforcing its resilience.

Overall, BYE generalizes the *concept of model-internal reaction* to poisoning from CNN saliency to Transformer entropy, and from local patches to global alignment disruptions—establishing a new paradigm for self-supervised backdoor purification.

# G   Broader Impact

This work offers a self-supervised defense mechanism that strengthens the safety of MLLMs against backdoor attacks in FTaaS scenarios. By detecting poisoned data without clean references, it helps reduce the risk of malicious model behaviors in critical applications such as education, healthcare, and autonomous systems. However, revealing that low attention entropy is a reliable signal for detecting poisoned samples may also motivate adversaries to craft more evasive triggers that diffuse attention or imitate benign entropy patterns. To mitigate such risks, we advocate for future research on adaptive defenses and the development of robust auditing tools for fine-tuning pipelines.

