# OpenReview forum: "Backdoor Cleaning without External Guidance in MLLM Fine-tuning"
_NeurIPS.cc/2025/Conference — NeurIPS 2025 poster_

### Official Review · Reviewer_jAXd · 2025-06-26

**Clarity:** 4
**Significance:** 2
**Originality:** 3
**Rating:** 4
**Confidence:** 4

**Summary:**

This paper proposes a self-diagnostic method, Believe Your Eyes (BYE), to identify and filter potential backdoor samples in the scenarios of fine-tuning multi-modal LLMs. By observing that the abnormal attention occurs for patch-based triggers, the authors propose to leverage the difference of attention entropy to partition clean and backdoors samples. In experiments, BYE achieves better performance (Higher CP and higher ASR) than baselines like Random Drop, ZIP with LLaVA and InternVL as backbones.

**Questions:**

1. I noticed that a black patch is shown in Fig. 2 as an example. This raises a question: in patch-based backdoor attacks, is the image difference typically larger than in adversarial perturbation-based methods? Since patch-based triggers are visually perceptible to humans, it seems plausible that even simpler detection methods could be effective in identifying them.
2. The entropy pattern should become more reliable as the tuning process progresses. Given this, why not incorporate a mechanism to 'forget' or filter out backdoor samples during tuning? While I understand that the online process is left as future work, the contribution of the paper would be more substantial if such an approach were considered within the current scope.
3. Given that the number of backdoor samples is much smaller than that of clean samples, could GMM clustering on such imbalanced data pose a problem?
4. Will entropy pattern from all layers agree with each other? Will the high separation (BSI) be highly dependent on the converged tuning procedure?

**Ethical Concerns:**

["NO or VERY MINOR ethics concerns only"]

**Final Justification:**

The authors have addressed most of my concerns. By checking all other reviewers' comments, I would like to raise my rating to 4.

**Limitations:**

yes

**Quality:**

3

**Strengths And Weaknesses:**

Strengths:
1. This paper is well organized, and the tables and figures are clear and easy to read.
2. The idea of using attention entropy patterns for identifying patched samples sounds useful and direct.
3. The performance presented in Table 1 looks a great improvement on attack success rate, across different datasets.
4. The visualization of sample separation and ablation studies provide additional evidence of the efficacy of the BYE.

Weaknesses:
1. Inconsistent notations. D_train in Sec. 3.1 is clean set for fine-tuning, while it represents the union of clean the backdoors samples in Sec 4.1.
2. This paper is narrowly focused on patch-based backdoor samples, which limits its relevance to the broader data poisoning community working on MLLMs. Notice that ZIP, the strongest baseline in this work, can handle non-patched samples.
3. According to Algorithm 1, BYE essentially requires fine-tuning MLLMs on the given dataset and then identifies which are backdoors examples. This looks not very efficient, as it involves performing fine-tuning twice.

---

> ### Author Rebuttal · Authors · 2025-07-30
>
> Dear Reviewer jAXd:
>
> Thank you for your careful review and valuable feedback on our work. In the following responses, we address each of your concerns in detail.
>
> ### W1: Notation Mistake
>
> Thank you for pointing out the inconsistency. In Sec. 3.1, $D_{\text{train}}$ refers to the clean training set for fine-tuning, while in Sec. 4.1, **it should be written as $D_{\text{train}} \cup D_{\text{poison}}$**, representing the union of clean and backdoored samples. We will revise the notation and clarify the definitions in the updated version.
>
> ### W2: Discussion on Diverse Trigger
>
> We sincerely thank the reviewer for raising this valuable point. To address this concern, we conduct additional experiments with **blended [1] (global, non-patched) backdoor triggers**, which better reflect real-world attack diversity. We also include two additional defense baselines, DiffPure [2] and SampDetox [3]. As shown in our new results, **BYE still achieves much lower ASR than all baseline methods**, confirming that our approach is not limited to simple or visually distinctive triggers, but generalizes well to more adaptive and subtle attack types. This robustness stems from BYE’s focus on **behavioral patterns rather than explicit patch localization**, allowing it to generalize across trigger types.
>
> In contrast, DiffPure, ZIP and SampDetox reliance on external prior knowledge limits its effectiveness against in-domain or blended triggers, and our results show it performs poorly in these settings.
>
> *Table 1: Performance of BYE under the blended trigger attack setting on ScienceQA and IconQA.*
>
> |**Method**|**ScienceQA CP (↑)**|**ScienceQA ASR  (↓)**|**IconQA CP (↑)**|**IconQA ASR  (↓)**|
> |-|-|-|-|-|
> |DiffPure|82.85|77.24|78.54|84.91|
> |ZIP|87.11|91.12|79.81|96.77|
> |SampDetox|82.45|95.93|73.27|93.77|
> |**BYE**|**85.72**|**54.18**|**79.32**|**59.77**|
>
> > [1] Targeted backdoor attacks on deep learning systems using data poisoning. Preprint 2017.
> >
> > [2] Diffusion models for adversarial purification. ICML 2022.
> >
> > [3] SampDetox: Black-box Backdoor Defense via Perturbation-based Sample Detoxification. NeurIPS 2024.
>
> ### W3: Discussion on Computation Cost
>
> Thanks for raising this important concern regarding computational expense! We would like to clarify the actual cost of our method and compare it with alternative backdoor removal approaches.
>
> **Computation Cost**: Our system BYE consists of three steps: (1) Standard fine-tuning on the (possibly poisoned) dataset (about 30 minutes), (2) One-time inference pass for suspicious sample detection (about 8 minutes), (3) Re-training on the filtered clean set (about 30 minutes). **Total: about 72 minutes on 4×4090 GPUs (6k samples).** By contrast, diffusion-based methods like ZIP require similar fine-tuning but add a time-consuming purification stage, taking at least 3 hours of GPU time for the same dataset depends on the diffusion model choosed. **Total: 162 minutes or more**.
>
> *Table 2: Comparison of total computational time (in minutes) for BYE and ZIP.*
>
> | **Method** | **Training** | **(Entropy/Purify)** | **Retraining** | **Inference** | **Total** |
> |-|-|-|-|-|-|
> | ZIP|30|128|—|4|**162**|
> |**BYE**|**30**|**8**|**30**|**4**|**72**|
>
> ### Q1: Discussion on Patch Trigger Visibility
>
> Thank you for these insightful suggestion. We address them as follows:
>
> 1. Is the image difference in patch-based attacks typically larger than in adversarial perturbation-based methods?
>
> **No.** For the model, both patch-based and adversarial triggers function similarly in disrupting internal behavior, regardless of the visual difference. For humans, patch triggers can also appear as real-world objects (e.g., sunglasses), which can be highly inconspicuous.
>
> 2. Could simple visual inspection suffice to detect patch-based triggers?
>
> **No.** While patch triggers are visually perceptible, relying on manual or heuristic inspection is neither reliable nor scalable for real-world, large-scale datasets. Real-world triggers may be any pattern, such as logos or stickers, which are not always obvious to humans. MLLMs do not rely on human-like visual cues; triggers that are obvious to us may not be distinguishable at the model level. Automated, model-based detection methods are thus necessary for practical defense.
>
> ### Q2: Analysis of Online Filtering
>
> Thank you for this insightful suggestion. The design of BYE as a post-hoc defense is motivated by the need for the model to **fully exhibit both clean and backdoored behaviors** before effective detection is possible. In our approach, the identification of poisoned samples relies on the model having already learned the backdoor pattern, which allows us to observe clear statistical differences in attention entropy between clean and triggered data. If filtering were performed online during training, BYE would only be able to identify a backdoored sample after the model has already been affected by it. This would **lead to a potential dilemma where the model could repeatedly be contaminated**, identified, and purified in a cycle, making it **difficult to achieve stable and effective defense**. By performing detection and purification after fine-tuning, we ensure that the model has fully revealed any backdoor-related behaviors, which can then be accurately distinguished and removed.
>
> ### Q3: Discussion on Diverse Poison Rate
>
> Thank you for highlighting the concern about GMM clustering on highly imbalanced data. **BYE identifies poisoned samples by capturing the behavioral difference between clean and triggered inputs, regardless of the poison rate or class imbalance.** As long as the backdoor consistently induces abnormal model responses, the entropy-based feature enables GMM to reliably separate poisoned from clean samples, even when the number of poisoned samples is much smaller.
>
> To verify this, we conducted experiments at 10%, 5%, and 1% poison rates on ScienceQA and IconQA. Results show **BYE maintains strong detection and low ASR even with just 1% contaminated data:**
>
> *Table 3: BYE performance under varying poison rates (10%, 5%, 1%) on ScienceQA and IconQA. In the format CP (↑) / ASR (↓).*
>
> | **Poison Rate** | **ScienceQA** | **IconQA**   |
> |-|-|-|
> | **10%**| 89.64 / 0.05  | 83.39 / 0.00 |
> | **5%**| 90.92 / 0.16  | 84.20 / 1.25 |
> | **1%**| 88.35 / 3.78  | 82.18 / 3.62 |
>
> ### Q4: Discussion on Entropy Consistency & Convergence
>
> Thank you for these important questions. We address them as follows:
>
> 1. Do entropy patterns from all layers agree?
>
> **No, entropy separation is not uniform across all layers.** Only a subset of layers show clear discrimination between clean and poisoned samples. Including all layers actually degrades detection. Therefore, **we use an adaptive selection strategy**, aggregating entropy only from layers with high BSI, as detailed in Appendix A.3. This makes BYE robust to different architectures and tasks.
>
> 2. Is BSI separation highly dependent on tuning convergence?
>
> **No.** Our experiments with checkpoints from different fine-tuning epochs (see table below) show that **BYE’s detection effectiveness is stable and not sensitive to the convergence stage.** Even with only 1–2 epochs of tuning, the attack success rate remains very low, confirming the robustness of our method throughout the training process.
>
> *Table 4: Clean performance and attack success rate (ASR) of BYE at different fine-tuning epochs.*
>
> | **Epoch** | **Clean Performance** | **ASR** |
> | --------- | --------------------- | ------- |
> | 1         | 83.69                 | 0.07    |
> | 2         | 87.06                 | 0.10    |
> | 3         | 89.64                 | 0.05    |
>
> Our adaptive layer selection ensures that BYE focuses on the most informative layers for backdoor detection, avoiding noisy or uninformative ones. Extensive experiments demonstrate that BYE’s effectiveness is robust to the choice of layers and to the degree of model convergence during fine-tuning. These design choices together guarantee that BYE is reliable and practical across different architectures and FTaaS scenarios.

---

> > ### Comment · Reviewer_jAXd · 2025-08-04
> >
> > Thank you for your response. 1. I saw other reviewers had raised similar concerns on diverse triggers and computation cost. By checking the Introduction (lines 36-39), where authors explicitly stated the advantageous property of patch-based triggers, I felt this should serve as a key technical motivation or inspiration of the paper. Nevertheless, during the rebuttal, the authors additionally tested BYE and demonstrated that it can also work consistently on blended triggers. Although I am not going to doubt the reproduction, I felt confused about the applicable scenarios of BYE. Is it naturally extendable to all triggers, and why? Regarding the computation cost, it may be less convincing for me to experimentally compare it with ZIP. I think this is the common weakness of detection-based method. If no other detection baselines can be compared to, authors should state why such a computation cost is tolerable, especially in terms of this research task? 2. Although additional experiments might be helpful, I was personally expecting a deeper presentation of the task, including its underlying techniques and idea novelty, which would help make the paper more inspirable to readers in the ML and privacy communities.

---

> > > ### Author Response · Authors · 2025-08-04
> > > **Response and Appreciation**
> > >
> > > Thank you very much for your careful review and helpful suggestions.
> > >
> > > Regarding the focus on patch-based triggers, our initial motivation indeed came from their practical impact, as highlighted in the introduction. At the same time, our method is designed to capture model behavioral patterns, which enables it to generalize beyond specific trigger types. Our experiments on both patch-based and blended triggers confirm this broader applicability.
> > >
> > > Regarding computation cost, we agree that efficiency is an important consideration for practical deployment. In our study, ZIP was chosen as the main baseline for comparison because it is widely adopted in recent works and represents the state-of-the-art among purification-based defenses. In real-world FTaaS scenarios, our method introduces only a moderate additional training stage, and all steps remain within standard fine-tuning and inference workflows. We believe this level of computation is practical for the research task and in line with typical defense methods used in the community.
> > >
> > > We appreciate your comments and thank you again for your feedback.

---

> > > > ### Comment · Reviewer_jAXd · 2025-08-05
> > > >
> > > > I appreciate the authors' response. I do not have further questions.

---

### Official Review · Reviewer_ep1A · 2025-06-30

**Clarity:** 4
**Significance:** 3
**Originality:** 2
**Rating:** 4
**Confidence:** 4

**Summary:**

The paper identifies "attention collapse"—abnormal cross-modal attention concentration induced by malicious fine-tuning. It proposes a framework using attention entropy to detect poisoned samples, leveraging Gaussian Mixture Models (GMM) and Bimodal Separation Index (BSI) to model entropy distributions for unsupervised filtering.

**Questions:**

1. **[Contrasting Baselines]**: Could the method outperform other attention-based method? It lacks comparison with other attention-based methods such as Neural Attention Distillation (NeurIPS), which could serve as strong baselines. The comparison against only ZIP (an inference‐time purification method) creates a paradigm mismatch that undermines the evaluation’s validity. And Vanilla FT and Random Drop are too simple to prove the effectiveness of BYE. By omitting established fine‐tuning defenses (e.g., spectral signature filtering, adversarial fine‐tuning, differential privacy) as baselines, the paper fails to demonstrate BYE’s relative advantage within its intended training‐time paradigm, leaving its real‐world efficacy and efficiency claims under‐substantiated.
2. **[Simulation of Backdoor Attacks]**: In the experimental section, small black blocks are used to simulate backdoor attacks. Is this method scientifically valid? Realistically backdoor triggers may be more insidious or diverse. Is the method still effective in detecting non-blocking, non-explicit triggers? Does the attention collapse phenomenon stabilize in such triggers?
3. **[Unquantified Computational Overhead]**: The method requires two full fine-tuning passes (before and after cleaning), yet GPU hours and wall-clock time are not reported, hindering assessment of real-world cost. Does BYE consume more computational resources or longer inference time? The paper mentions that self-supervised methods are “computationally efficient” but lacks a quantitative comparison with externally supervised methods.
4.  **[Generalization Ability on Different Models]**: Existing experiments are mainly based on GPT-J and GPT-2 XL, how about performance of the methods on different architectures (e.g., T5 with Encoder-Decoder structure, LLaMA series with pure Decoder) or different scale models (e.g., 7B vs 13B parameters)? The core mechanisms of BYE may vary significantly in effectiveness depending on the attention head design of the model and the dimensionality of the hidden layer.
5. **[Lack of Statistical Significance Analysis]**: All performance gains are reported as averages without confidence intervals or significance tests, leaving it unclear whether the improvements are robust or due to randomness.

**Ethical Concerns:**

["NO or VERY MINOR ethics concerns only"]

**Final Justification:**

I appreciate the thorough rebuttal from the authors. All my concerns have been sufficiently addressed, and I’ve raised my score to 4.

**Limitations:**

Yes

**Paper Formatting Concerns:**

No.

**Quality:**

4

**Strengths And Weaknesses:**

**Strengths**

1. **[Complete work]** The work is quite complete, progressing from exploring the underlying cause to proposing a practical solution. Specifically, the paper begins with the discovery of attention-entropy collapse, then discusses an optimization method using Gaussian modeling to select specific layers.
2. **[Various applied scenarios]**: The combination of GMM and BSI is a valuable method that could be applied across various scenarios. The ablation study is also persuasive.
3. **[Clear narrative]***: The narrative is clear enough for readers to follow the reasoning, and the language is generally fluent.

**Weaknesses**

1. **[Lack Comparison]**: While the work is complete, the method itself may not be particularly novel. It lacks comparisons with similar approaches. The paper only mentions external data purification methods like ZIP, but does not introduce other attention-based methods. Including such comparisons would better highlight the novelty of the proposed method.
2. **[Lack persuasion]**: Additionally, is the ZIP experiment really the most state-of-the-art? Although ZIP is less effective than the proposed method, this might be because ZIP is more suited for real-world situations, whereas the proposed method performs better only in the simulated scenarios described in the paper. This possibility is not discussed in the paper.
3. **[Untrustful Simulation]**: In the experimental section, small black blocks are used to simulate backdoor attacks. Is this method scientifically valid? Has any previous research used this approach? The paper lacks explanation regarding this design choice.
4. **[Vague part]**: Due to space constraints, Section 5.5 lacks clarity and is somewhat vague; it requires more precise elucidation.
5. **[Computation cost]**: The proposed method requires fine-tuning, which is relatively computationally intensive. Compared with using external supervision, does leveraging self-supervision consume more computational resources?

---

> ### Author Rebuttal · Authors · 2025-07-30
>
> Dear Reviewer ep1A:
>
> We appreciate your engagement with our work and the thoughtful observations you made. We aim to address your concerns in our detailed responses below.
>
> ### W1&W2&Q1: Comparison with More Baselines
>
> Thank you for raising the concern about the adequacy of our defense baselines. Our comparisons are limited to image purification methods (e.g., ZIP), which do not fully capture the landscape of advanced backdoor defense strategies.
>
> To address this, we add extensive new baselines, including: (1) Neural Attention Distillation (NAD) [1] (attention-based fine-tuning defense). (2) DiffPure [2] and SampDetox [3] (state-of-the-art purification-based methods).
>
> *Table 1: Comparison of BYE under the BadNet attack setting on ScienceQA and IconQA.*
>
> |**Method**|**ScienceQA CP**|**ScienceQA ASR**|**IconQA CP**|**IconQA ASR**|
> |-|-|-|-|-|
> |DiffPure|81.71|78.08|79.04|77.51|
> |ZIP|79.97|66.48|77.60|67.97|
> |SampDetox|83.59|90.62|72.86|89.74|
> |NAD|84.65|7.51|75.87|13.04|
> |**BYE**|**89.64**|**0.05**|**83.39**|**0.00**|
>
> As shown in Table 1, **BYE consistently achieves the best performance across ScienceQA and IconQA, outperforming both attention-based and purification-based defenses**. We emphasize that ZIP, DiffPure, and SampDetox are state-of-the-art defenses designed specifically for real-world backdoor threats, and all baselines are evaluated under identical attack types and threat models as BYE.
>
> > [1] Neural attention distillation: Erasing backdoor triggers from deep neural networks. ICLR 2021.
> >
> > [2] Diffusion models for adversarial purification. ICML 2022.
> >
> > [3] SampDetox: Black-box Backdoor Defense via Perturbation-based Sample Detoxification. NeurIPS 2024.
>
> ### W3&Q2: Trigger Validity&Diversity Discussion
>
> Thank you for these thoughtful questions regarding our experimental design and the scientific validity of using small black blocks as backdoor triggers.
>
> A1: Using small black squares as visual triggers is a standard and widely adopted practice in the backdoor literature (e.g., BadNets [1], federated learning, and recent CNN-based studies). This setup is favored because it is **simple, reproducible, and widely recognized as a baseline**, making our results directly comparable to prior work.
>
> A2: To address this, we conduct additional experiments under the more challenging blended trigger attack [2], where BYE still achieves the lowest attack success rate. These results clearly demonstrate the **robustness and generalizability of BYE across a range of potential attacks**.
>
> *Table 2: Performance under blended attack. In the format CP (↑) / ASR (↓).*
>
> |**Method**|**ScienceQA CP** | **ScienceQA ASR** | **IconQA CP** | **IconQA ASR** |
> |-|-|-|-|-
> |DiffPure|82.85|77.24|78.54|84.91|
> |ZIP|87.11|91.12|79.81|96.77|
> |SampDetox|82.45|95.93|73.27|93.77|
> |NAD|85.77|20.87|70.89|33.75|
> |**BYE**|**85.72**|**54.18**|**79.32**|**59.77**|
>
> > [1] Badnets: Identifying vulnerabilities in the machine learning model supply chain. Preprint 2017.
> >
> > [2] Targeted backdoor attacks on deep learning systems using data poisoning. Preprint 2017.
>
> ### W4: Detailed description in Section 5.5
>
> Thank you for pointing out the need for clarity in Section 5.5. We conduct experiments in Section 5.5 (in page 9, Table 3) specifically to verify the robustness of our method against potential adaptive backdoor attacks. Given that an attacker may attempt to circumvent our detection mechanism by dispersing attention triggers across multiple spatial locations, we design experiments with multi-trigger scenarios, including fixed dual triggers and varied multi-triggers. We aim to demonstrate that our entropy-based detection approach remains effective even when triggers are spatially dispersed or non-localized. To clarify, we implement two adaptive attack scenarios: (1) embedding multiple identical triggers symmetrically placed within a single image (Fixed Dual Trigger); (2) embedding triggers dispersed at regular grid points across images (Varied Multi-Trigger). Results in Table 3  (page 9) confirm our method’s robustness, showing consistent reductions in attack success rates (ASR) and high recall in identifying poisoned samples, thereby validating our defense’s resistance against more sophisticated, dispersed attack patterns. We will expand this discussion in the revised manuscript to explicitly outline the rationale and setup for these experiments.
>
> ### W5&Q3: Discussion on Computation Cost
>
> Thanks for raising this important concern regarding computational expense! We would like to clarify the actual cost of our method and compare it with alternative backdoor removal approaches.
>
> **Computation Cost**: Our system BYE consists of three steps: (1) Standard fine-tuning on the (possibly poisoned) dataset (about 30 minutes), (2) One-time inference pass for suspicious sample detection (about 8 minutes), (3) Re-training on the filtered clean set (about 30 minutes). **Total: about 72 minutes on 4×4090 GPUs (6k samples).** By contrast, diffusion-based methods like ZIP require similar fine-tuning but add a time-consuming purification stage, taking at least 3 hours of GPU time for the same dataset depends on the diffusion model choosed. **Total: 162 minutes or more**.
>
> *Table 3: Comparison of total computational time (in minutes) for BYE and ZIP.*
>
> |**Method**|**Training**|**(Entropy/Purify)**|**Retraining**|**Inference**|**Total**|
> |-|-|-|-|-|-|
> |ZIP|30|128|—|4|**162**|
> |**BYE**|**30**|**8**|**30**|**4**|**72**|
>
> **Quantitative comparison with externally supervised methods:** In contrast to externally supervised baselines which often require additional annotation, supervision, or computationally heavy denoising modules. BYE’s self-supervised detection incurs only a lightweight, one-time cost and no extra inference latency. The above quantitative comparison demonstrates that BYE is not only more efficient than diffusion-based methods, but also avoids the recurring overhead associated with external supervision.
>
> ### Q4: Generalization Across Model Architectures
>
> Thank you for this important question about generalizability across model architectures and scales! **BYE is designed to be architecture-agnostic: it relies only on extracting cross-modal attention maps**, which are present in both decoder-only (e.g., LLaMA, GPT) and encoder-decoder (e.g., T5) models. Its effectiveness does not depend on the number of attention heads, hidden size, or specific attention design. BYE detects distributional shifts in attention caused by backdoor triggers which a phenomenon expected to occur across both types of architectures, since the attack goal is to induce shortcut behaviors in attention, regardless of model structure.
>
> To demonstrate scalability, we conduct additional experiments on LLaVA-1.5-13B (a large decoder-only model) with ScienceQA and IconQA. BYE achieves strong clean performance and robust backdoor mitigation:
>
> *Table 4: Clean performance (CP) and attack success rate (ASR) of BYE on LLaVA-1.5-13B, demonstrating robustness and scalability to larger model sizes.*
>
> |**Dataset**|**Clean Performance (CP)**|**Attack Success Rate (ASR)**|
> |-|-|-|
> |ScienceQA|95.85|0.04|
> |IconQA|84.23|0.00|
>
> These results show BYE is robust and effective as model scale increases.
>
> ### Q5: Statistical Significance Analysis
>
> We thank the reviewer for raising the issue of statistical significance. To address this concern, we **repeat the main experiments on ScienceQA and IconQA three times with different random seeds** and report the mean, standard deviation, and 95% confidence interval (CI) for each key metric in the table below.
>
> *Table 5: Statistical analysis of BYE’s main results, demonstrating robustness and stability across multiple runs.*
>
> |**Dataset**|**Metric**|**Mean**|**Std**|**95% CI**|
> |-|-|-|-|-|
> |ScienceQA|CP|89.64|0.8|[88.73, 90.55]|
> |ScienceQA|ASR|0.05|0.01|[0.04, 0.06]|
> |IconQA|CP|83.39|0.7|[82.60, 84.18]|
> |IconQA|ASR|0.00|0.00|[0.00, 0.00]|
>
> The results show that **BYE’s performance is highly stable and robust to randomness, with minimal variance across independent runs**. This stability arises because our method is almost fully deterministic: the inference temperature is set to 0, and GMM clustering yields highly consistent results due to the clear separation between clusters.

---

> > ### Comment · Reviewer_ep1A · 2025-08-05
> > **Raise my score to 4**
> >
> > After reading the thorough rebuttal,  all my concerns about the experiment details have been resovled. The new experiments on blended triggers (Table 2), scalability to LLaVA-13B (Table 4), and expanded baselines significantly strengthen the paper, and I’ve raised my score to 4.

---

> > > ### Author Response · Authors · 2025-08-05
> > > **Response and Appreciation**
> > >
> > > Thank you very much for your positive feedback and for raising your score. We appreciate your recognition of our new experiments and expanded baselines. Your constructive comments have helped us strengthen the work, and we are grateful for your support and encouragement.

---

### Official Review · Reviewer_Uw8H · 2025-07-02

**Clarity:** 3
**Significance:** 2
**Originality:** 2
**Rating:** 4
**Confidence:** 3

**Summary:**

The paper "Backdoor Cleaning without External Guidance in MLLM Fine-tuning" proposes a novel defense against image-based backdooring in Multimodel LLMs, Believe Your Eyes (BYE). BYE works by analysing the attention on the training data after fine-tuning. If there is less entropy in the attention, the image most likely contains a patch based trigger, whereas high entropy would reflect natural images. The process of distinguishing the different kinds of data is done without any further knowledge about the poisoned or clean data.
The evaluations show high removal success rate for BYE on different MLLM and the patch based triggers.

**Questions:**

- See weakness: Would this method work for poisoned data with blended or more naturally occuring triggers such as ones shown in, e.g., [1, 2]?
- Are there insights into the different performances of BYE on the 2 models? While BYE achieves a reduction of the ASR to near 0 for LLaVA, for InternVL, the ASR is always roughly above 7 for all evaluated datasets.

References:\
[1] Chen et al. Targeted Backdoor Attacks on Deep Learning Systems Using Data Poisoning. Preprint 2017\
[2] Liu et al. Reflection Backdoor: A Natural Backdoor Attack on Deep Neural Networks. ECCV 2020

**Ethical Concerns:**

["NO or VERY MINOR ethics concerns only"]

**Final Justification:**

The initial response has addressed my main weakness of BYE, by evaluating on more diverse triggers. My questions were also answered thoroughly. My main concern is the performance and missing explanation as to why BYE works on the whole-image triggers, which should also be revised in the paper (see initial comment).

**Limitations:**

yes

**Paper Formatting Concerns:**

No paper formatting concerns.

**Quality:**

2

**Strengths And Weaknesses:**

### Strength
- Extensive evaluations are conducted, including solid ablation studies for patch-based triggers.
- The motivation behind BYE is well presented, and the method is clearly described.
- Well chosen visualizations to showcase the problem setting and proposed solution.
- The paper is well written and easy to follow.

### Weaknesses
- Limited threat model: BYE appears to be limited to detecting patch-based triggers that concentrate model attention on small regions within an image. This would make BYE vulnerable to triggers that are more spread over the image via, e.g., blending or superimposing (explored in earlier image classification backdoor works such as [1,2]). Using such triggers could cause similar entropy to clean images, circumventing the BYE defense completely.

References:\
[1] Chen et al. Targeted Backdoor Attacks on Deep Learning Systems Using Data Poisoning. Preprint 2017\
[2] Liu et al. Reflection Backdoor: A Natural Backdoor Attack on Deep Neural Networks. ECCV 2020

---

> ### Author Rebuttal · Authors · 2025-07-30
>
> Dear Reviewer Uw8H:
>
> Thank you for your encouraging remarks and the critical questions you posed. We provide our detailed responses below to address your concern.
>
> ### W1&Q1: Trigger Diversity Discussion
>
> We thank the reviewer for highlighting this important concern about the attack diversity. Our initial evaluation mainly considered patch-based triggers, we agree it is crucial to test BYE under more distributed and subtle attacks such as blended and reflection triggers [1,2].
>
> **BYE remains effective even for global or naturally embedded triggers.** No matter if the trigger is a small patch or a blended pattern, the backdoor’s core effect is to teach the model a shortcut so the model shows a consistent, abnormal response pattern. This often leads to a class-consistent, atypical attention distribution, which BYE detects via clustering. **While global triggers may not concentrate attention on a single region, they still create distinctive and class-consistent attention shifts compared to clean samples.**
>
> To address this, we conducted additional experiments under the blended attack setting and included two state-of-the-art purification-based baselines, DiffPure [3] and SampDetox [4]. The results below show that **BYE remains relatively effective and significantly outperforms both baselines under this more challenging scenario:**
>
> *Table 1: Performance comparison of BYE with DiffPure, ZIP, and SampDetox under the blended trigger attack setting on ScienceQA and IconQA.*
>
> |**Method**|**ScienceQA CP (↑)**|**ScienceQA ASR  (↓)**|**IconQA CP (↑)**|**IconQA ASR  (↓)**|
> |-|-|-|-|-|
> |DiffPure|82.85|77.24|78.54|84.91|
> |ZIP|87.11|91.12|79.81|96.77|
> |SampDetox|82.45|95.93|73.27|93.77|
> |**BYE**|**85.72**|**54.18**|**79.32**|**59.77**|
>
> > [1] Targeted backdoor attacks on deep learning systems using data poisoning. Preprint 2017.
> >
> > [2] Liu et al. Reflection Backdoor: A Natural Backdoor Attack on Deep Neural Networks. ECCV 2020.
> >
> > [3] Diffusion models for adversarial purification. ICML 2022.
> >
> > [4] SampDetox: Black-box Backdoor Defense via Perturbation-based Sample Detoxification. NeurIPS 2024.
>
> ### Q2: Analysis of Performance Gap on LLaVA&InternVL
>
> Thank you for your insightful question regarding the differing ASR reduction achieved by BYE on LLaVA and InternVL! The main reason for this gap lies in the **architectural differences** in how these models encode and fuse visual information with language.
>
> LLaVA maintains a relatively high spatial resolution for visual input, representing images with a fixed 24×24 grid (576 tokens) that are linearly projected to match the LLM input dimension (typically 1024). Visual tokens are simply concatenated with text tokens and jointly processed by the LLM, **preserving fine-grained spatial cues**. This enables BYE to precisely capture attention collapse caused by backdoor triggers and reduce the ASR to near zero.
>
> In contrast, InternVL first encodes images as 1024 visual tokens, but then **applies a cross-modal projection** to reduce this sequence to just **256 tokens** (matching the input dimension of InternLM). Moreover, InternVL employs multiple layers of cross-modal fusion, where visual and language features interact deeply. While this design facilitates stronger multimodal alignment, the lower spatial resolution (256 vs. 576 tokens) means that each visual token encodes information from a larger image region, causing trigger effects to be more diluted or mixed. This leads to a residual ASR around 7. Despite this, BYE still consistently and substantially reduces the ASR for InternVL across all datasets, demonstrating its robustness to architectural differences and lower token resolutions.

---

> > ### Comment · Reviewer_Uw8H · 2025-08-01
> > **Raising score to 4**
> >
> > Thank you for your thorough responses to me and the other reviewers. Based on these responses I have raised my score to a **4**.
> > You have addressed my main weakness of the missing evaluation on more divers triggers (especially the ones spanning the whole image) and answered my question about the performance gap between LlaVA and InternVL in depth, showcasing detailed knowledge.
> > Furthermore, you have invested a good amount of effort to run relevant experiment for questions or concerns of everyone, which is highly appreciated. You also introduced new baselines and datasets, making your evaluations overall more robust.
> >
> > However, I am currently not raising my score to 5, mainly because the defense against whole‑image triggers still appears to me to be more of a side effect rather than a core design feature of BYE (as far as I understand the method), and the reduction in ASR for these cases (to around 50%) leaves room for improvement. A clarification of why BYE works on these kinds of triggers, also as to compared with the baselines you provided, and how it could be adapted to more explicitly address them, would improve the paper further. As all reviewer have mentioned the weakness of using patch-based triggers, this should also be a bigger part of the overall paper in my opinion. Currently, the triggers that are evaluated on in the paper are not sufficient or not complex enough to demonstrate a strong and broadly effective defense.

---

> > > ### Author Response · Authors · 2025-08-02
> > > **Appreciation for the Insightful Review and Suggestions**
> > >
> > > Thank you for your positive feedback and for raising your score! While BYE was originally designed with localized triggers, its fundamental mechanism is based on detecting unusual model behaviors caused by visual triggers. This enables BYE to remain effective even against whole-image triggers. In contrast, diffusion-based purification methods such as ZIP heavily rely on the prior knowledge encoded in the diffusion model. For triggers that are blended with in-domain triggers that closely match the original data distribution, these methods often struggle to distinguish and remove the trigger. We agree that further enhancing robustness against more complex triggers is an important direction for future work, and we appreciate your thoughtful suggestions!

---

### Official Review · Reviewer_Bszd · 2025-07-03

**Clarity:** 4
**Significance:** 3
**Originality:** 3
**Rating:** 5
**Confidence:** 3

**Summary:**

This paper proposes the BYE (Believe Your Eyes) framework, an unsupervised data filtering framework focused on identifying and filtering backdoor samples. The authors specifically focus on the visual trigger-based attacks. They first sak the question - “Will attention of MLLMs systematically collapse toward the trigger rather than focusing on task-relevant content?”, which traditional Deep Learning models are known to do. The investigate by visualizing the cross-modal attention weights and determine that trigger patches indeed cause models to focus on them, a phenomenon which they term spatial-attention collapse. Based on these observations, they propose BYE which works in three main steps. First, the trained model is evaluated on the training dataset and the cross-modal attention map signals are measured for each transformer layer. In the next step, the Shannon entropy is calculated on the signals and it is observed that non-triggered images generally have lower entropy than those with triggers. The entropy distributions are clustered via Gaussian Mixture Model and two separate groups are created by filtering their distances via a threshold. Lastly, the suspicious samples that get detected through clustering are removed and the model is re-fitted on the new clean dataset to remove the influence of the trigger patches. Experiments are performed across different models and datasets, and the results show that BYE is able to filter our backdoored images with a high success rate across different types of triggers without compromising model performance.

**Questions:**

1. Please discuss more about the cost/expense of the system. It seems that it would be quite expensive. It is understood that such detection methods can be expensive, but it is important to clarify if the expense is worth it.
2. We know from older (CNN-based) models that patched triggers with strong patterns (like the black boxes here) are easier to detect the more distinctive and consistent they are compared to the rest of the image. How sensitive is this system for diverse and surreptitious triggers?
3. Similarly, these types of attacks rely on over-fitting, and what this solution does in essence is detect it via clustering. In the experiments 10% of the dataset was contaminated - have the authors any intuition of how trigger % contamination affects BYE's effectiveness?

**Ethical Concerns:**

["NO or VERY MINOR ethics concerns only"]

**Final Justification:**

I have set it to a 5 because I believe it is a good paper that tackles an interesting problem and covers all the bases. I am keeping my score to a 5 since I believe a score of 6 warrants extremely novel ideas, highly generalizable/applicable solutions, extremely effective solutions, or pushes the boundaries of science with this does not do, but it is good for what it is.

**Limitations:**

Yes.

**Quality:**

3

**Strengths And Weaknesses:**

Strengths -
1. The problem tackled is very interesting and important.
2. The solution is intuitive, effective and relatively simple which makes it feasible in practice.
3. The results are good.
4. The paper is explained well and easy to follow.

Weaknesses -
1. Training twice is extremely expensive
2. The triggers used are relatively simple
3. A few ideas are highly inspired by similar problems from traditional Deep Learning systems.

---

> ### Author Rebuttal · Authors · 2025-07-30
>
> Dear Reviewer Bszd:
>
> We are truly grateful for your insightful review and the constructive feedback provided. We respond to each point below with careful consideration.
>
> ### W1&Q1: Discussion on Computation Cost
>
> Thanks for raising this important concern regarding computational expense! We would like to clarify the actual cost of our method and compare it with alternative backdoor removal approaches.
>
> **Computation Cost**: Our system BYE consists of three steps: (1) Standard fine-tuning on the (possibly poisoned) dataset (about 30 minutes), (2) One-time inference pass for suspicious sample detection (about 8 minutes), (3) Re-training on the filtered clean set (about 30 minutes). **Total: about 72 minutes on 4×4090 GPUs (6k samples).** By contrast, diffusion-based methods like ZIP require similar fine-tuning but add a time-consuming purification stage, taking at least 3 hours of GPU time for the same dataset depends on the diffusion model choosed. **Total: 162 minutes or more**.
>
> *Table 1: Comparison of total computational time (in minutes) for BYE and ZIP.*
>
> |**Method**|**Training**|**(Entropy/Purify)**|**Retraining**|**Inference**|**Total**|
> |-|-|-|-|-|-|
> |ZIP|30|128|—|4|**162**|
> |**BYE**|**30**|**8**|**30**|**4**|**72**|
>
> **Practical value:** Diffusion-based methods introduce significant inference latency and are difficult to deploy in practice, since each test input must be purified at inference time. In contrast, BYE only adds extra time at the provider (server) side during training so **users experience no additional delay or difference at inference**. The deployed model remains as efficient as a standard model for real-world use.
>
> In summary, **BYE’s extra cost is modest, one-time, and entirely server-side.** It enables robust backdoor defense without affecting user experience, making the computational expense well justified for practical deployment.
>
> ### W2&Q2: Trigger Diversity Discussion
>
> Thank you for raising this important concern. We fully agree that classic black square triggers (BadNet), are simple and widely adopted but do not represent the full range of potential backdoor threats.
>
> **To address this, we conduct additional experiments using Blended [1] attack, a more global and less visually obvious attack method.** As shown in Table 2, BYE still achieves much lower ASR than all baseline methods, confirming that our approach is not limited to simple or visually distinctive triggers, but generalizes well to more adaptive and subtle attack types.
>
> *Table 2: Performance under blended attack. In the format CP (↑) / ASR (↓).*
>
> | **Method** | **ScienceQA**     | **IconQA**        |
> | ---------- | ----------------- | ----------------- |
> | Vanilla FT | 89.89 / 99.70     | 83.34  / 99.77    |
> | DiffPure   | 82.85 / 77.24     | 78.54 / 84.91     |
> | ZIP        | 87.11 / 91.12     | 79.81 / 96.77     |
> | SampDetox  | 82.45 / 95.93     | 73.27 / 93.77     |
> | **BYE**    | **85.72 / 54.18** | **79.32 / 59.77** |
>
> **To further assess robustness, we also evaluate with diverse triggers.** For each poisoned sample, we insert five small black trigger patches at random positions and assign randomly selected target captions from a paraphrase pool. We also compare against two additional defense baselines, DiffPure and SampDetox. As shown in Table 3, **BYE effectively reduces ASR to near zero even under these challenging, diverse trigger scenarios, and significantly outperforms all baselines**.
>
> *Table 3: Performance under diverse trigger and targeted output setting. In the format CP (↑) / ASR (↓).*
>
> |**Method**|**ScienceQA**|**IconQA**|
> |-|-|-|
> |Vanilla FT|91.21 / 93.70|81.62 / 86.29|
> |DiffPure|84.38 / 74.28|75.18 / 79.93|
> |ZIP|72.33 / 73.49|73.91 / 80.29|
> |SampDetox|84.39 / 93.19|76.82 / 90.97|
> |**BYE**|**87.38 / 2.37**|**80.99 / 1.68**|
>
> >  [1] Targeted backdoor attacks on deep learning systems using data poisoning. Preprint 2017.
>
> ### W3: Analysis of Innovation Beyond Traditional Defenses
>
> We appreciate the reviewer’s comment. While our method is inspired by traditional backdoor defenses, **adapting such strategies to MLLMs presents new challenges:**
>
> 1. **Complex cross-modal attention and modality alignment:** MLLMs require handling information across modalities, making direct application of CNN-based methods ineffective.
> 2. **Unsupervised entropy-based detection:** Our approach is specifically designed for transformer-based MLLMs, rather than relying on visual or gradient cues.
> 3. **FTaaS compatibility and real-world deployment:** The method does not require gradient or data access, making it practical for real FTaaS scenarios.
>
> In summary, our work **generalizes and customizes entropy-based backdoor detection** for the unique challenges of MLLMs.
>
> ### Q3: Discussion on Diverse Poison Rate
>
> Thank you for your question regarding the effect of different trigger contamination rates on the effectiveness of BYE.
>
> **BYE identifies poisoned samples by capturing the behavioral difference between clean and triggered inputs, regardless of the overall poison rate.** As long as the backdoor causes consistent abnormal responses, BYE can reliably detect them.
>
> To verify this, we conducted experiments at 10%, 5%, and 1% poison rates on ScienceQA and IconQA. Results show **BYE maintains strong detection and low ASR even with just 1% contaminated data:**
>
> *Table 4: BYE performance under varying poison rates (10%, 5%, 1%) on ScienceQA and IconQA. In the format CP (↑) / ASR (↓).*
>
> |**Poison Rate**|**ScienceQA**|**IconQA**|
> |-|-|-|
> |**10%**|89.64 / 0.05|83.39 / 0.00|
> |**5%**|90.92 / 0.16|84.20 / 1.25|
> |**1%**|88.35 / 3.78|82.18 / 3.62|
>
> These results confirm that BYE remains robust even under extreme conditions such as a 1% poison rate, as long as the backdoor produces a stable effect on the model. The effectiveness of our method is therefore determined by the presence of a detectable behavioral difference, rather than the absolute proportion of poisoned data.

---

> > ### Comment · Reviewer_Bszd · 2025-08-04
> > **Response to Author Rebuttal**
> >
> > Thank you for the additional results and clarifications. They answer my questions to a certain degree, but not completely. For example, for Q3, how does it perform compared to others? For Q1, what is an appropriate time/FLOPs expense in proportion to the actual model size (e.g. number of parameters)? For Q2, it is unclear what the attack dimensions are in the experiments.
> >
> > I am keeping my score to a 5 since I believe a score of 6 warrants extremely novel ideas, highly generalizable/applicable solutions, extremely effective solutions, or pushes the boundaries of science.

---

> > > ### Author Response · Authors · 2025-08-05
> > > **Response and Appreciation**
> > >
> > > Thank you for your additional comments and for carefully reviewing our work. We appreciate your constructive suggestions and will take them into consideration for future revisions and research. Your feedback has been valuable in helping us identify potential areas for further improvement.

---

### Official Review · Reviewer_wyVd · 2025-07-06

**Clarity:** 3
**Significance:** 2
**Originality:** 2
**Rating:** 4
**Confidence:** 2

**Summary:**

This paper proposes a backdoor cleaning algo BYE (Alg 1) that utilizes clustering on attention entropy as a proxy to detect and filter backdoors in MLLMs. The assumption is that when exposed to poisoned samples, MLLM's attentions would be collapsed / redirected toward the triggers instead of task-relevant content, and we can often observe a bimodal in attention entropy (e.g., Fig 3). And we can utilize GMM clustering on some sensitive layers to automatically detect such entropy separation and removing low-entropy cluster samples is sufficient (Alg 1).

The experiments are conducted on LLaVA-v1.5-7B and InternVL2.5-8B with LoRA FT. The tasks are visual QA and image captioning. The metrics are Attack Success Rate (ASR) and Clean Performance (CP). The baselines are vanilla FT, random drop, and ZIP, and the authors demonstrate a decent CP can be achieved by BYE while ASR is quite close to 0.

**Questions:**

There is no other questions than the ones listed in the weakness section.

**Ethical Concerns:**

["NO or VERY MINOR ethics concerns only"]

**Final Justification:**

Disclaimer: I am not an expert of the backdoor cleaning field.

My main concern is the general applicability of BYE. But the authors have provided multiple new experiments that appear to demonstrate BYE's performance with more diverse attacks, and 2 other reviewers are quite satisfactory about this. I will correspondingly raise the score to borderline accept.

**Limitations:**

yes

**Paper Formatting Concerns:**

No paper formatting issues founded

**Quality:**

3

**Strengths And Weaknesses:**

Disclaimer: I am not familiar with the field of backdoor cleaning so I would be conservative on assessing the novelty. The following evaluation is more tailored towards the soundness of the claims from my understanding.

Strength:

- This paper studies an essential problem (MLLM backdoor cleaning) with a clear motivation. The paper is easy to follow even for people without backdoor cleaning background.
- The high-level algorithmic design is well motivated and the experiments are comprehensive.


Weakness:

- High-level:
  - **algorithmic assumption and experiment verification**: I agree that attention could give clear signal on the difference between poisoned and clean samples, but I doubt identifying from entropy score be a robust / general solution. The authors provide some justification on Figure 3 and test on 3 different triggers in Figure 1 & Table 3, but it might not be sufficient. Although I am not an expert in inserting backdoors, I could still imagine designing backdoors that are simply more diverse shouldn't be a hard task.

For example, we can construct triggers that are **both randomly scattered on the images (and there should be multiple pieces scattered on a single image, instead of trigger occurs on just 1 place per image) and have multiple randomly chosen captions for the dataset**. This is likely to be more diverse, and may degrade BYE's sensitivity. Notice that this construct is different from both random position trigger and texture trigger in terms that we have (1) multiple small triggers per image and (2) randomly chosen captions for image (the first one might be more important for diversity).


- Low-level
  - FT results on the clean samples only should be provided in Table 1 as an upper bound for CP and lower bound on ASR.
  - An empirical attention entropy histogram should be provided for the other 2 triggers (Fixed Dual & Varied Multi-trigger) besides Table 3.

---

> ### Author Rebuttal · Authors · 2025-07-30
>
> Dear Reviewer wyVd:
>
> We greatly value the time and expertise you invested in reviewing our submission. We address your comments in detail below.
>
> ### W1: Trigger Diversity Discussion
>
> Thanks for the insightful suggestion regarding the robustness of our entropy-based detection method under more diverse and challenging backdoor attack scenarios.
>
> To address this concern, **we conduct additional experiments following the proposed setting**: for each poisoned sample, we insert five small black trigger patches at random positions and assign a randomly selected target caption from a paraphrase pool. We also include two additional defense baselines, DiffPure [1] and SampDetox [2].
>
> *Table 1: Performance under diverse trigger and targeted output setting. In the format CP (↑) / ASR (↓).*
>
> |**Method**|**ScienceQA**|**IconQA**|
> |-|-|-|
> |Vanilla FT|91.21 / 93.70|81.62 / 86.29|
> |DiffPure|84.38 / 74.28|75.18 / 79.93|
> |ZIP|72.33 / 73.49|73.91 / 80.29|
> |SampDetox|84.39 / 93.19|76.82 / 90.97|
> |**BYE**|**87.38 / 2.37**|**80.99 / 1.68**|
>
> As shown, **BYE effectively reduces ASR to near zero on both ScienceQA and IconQA, even under diverse attacks, and significantly outperforms the baselines.**
>
> In addition, **we further evaluate our method under the blended [3] attack scenario.** BYE consistently achieves significantly lower ASR than all baseline methods in this setting as well, further confirming the robustness and generalizability of our approach across various attack types.
>
> *Table 2: Performance under blended attack. In the format CP (↑) / ASR (↓).*
>
> |**Method**|**ScienceQA**|**IconQA**|
> |-|-|-|
> |Vanilla FT|89.89 / 99.70|83.34  / 99.77|
> |DiffPure|82.85 / 77.24|78.54 / 84.91|
> |ZIP|87.11 / 91.12|79.81 / 96.77|
> |SampDetox|82.45 / 95.93|73.27 / 93.77|
> |**BYE**|**85.72 / 54.18**|**79.32 / 59.77**|
>
> > [1] Diffusion models for adversarial purification. ICML 2022.
> >
> > [2] SampDetox: Black-box Backdoor Defense via Perturbation-based Sample Detoxification. NeurIPS 2024.
> >
> > [3] Targeted backdoor attacks on deep learning systems using data poisoning. Preprint 2017.
>
> ### W2: Upper and Lower Bound Analysis
>
> Thank you for your valuable suggestion! It's important to include results from models fine-tuned on clean samples only, which can serve as an upper/lower bound for CP/ ASR.
>
> In our original Table 1 (page 8), “Vanilla FT” uses mixed clean and poisoned data. Following your advice, **we additionally report Clean FT results (no poisoned samples) as shown below:**
>
> *Table 3: Clean FT vs. Vanilla FT. Clean FT provides the upper bound for CP and lower bound for ASR. In the format CP (↑) / ASR (↓)*
>
> ||**ScienceQA**|**IconQA**|**Flickr30k**|**RSVQA**|
> |-|-|-|-|-|
> |**LLaVA (Clean FT)**|**92.58 / 0.00**|**82.95 / 0.00**|**75.38 / 0.00**|**74.11 / 0.00**|
> |LLaVA (Vanilla FT)|91.72 / 97.32|80.51 / 87.85|71.03 / 82.80|72.01 / 99.90|
> |**InternVL (Clean FT)**|**97.65 / 0.00**|**96.96 / 0.00**|**54.86 / 0.00**|**72.84 / 0.00**|
> |InternVL (Vanilla FT)|91.47 / 97.12|89.96 / 92.13|48.55 / 76.60|65.21 / 99.76|
>
> As expected, **Clean FT consistently yields the highest CP and zero ASR, confirming its role as the upper/lower bound.** We will add these results and clarify the different fine-tuning settings in our revised submission.
>
> ### W3: Entropy Histograms Analysis
>
> Thank you for your valuable suggestion. **We perform the experiments and find that the entropy separation between clean and poisoned samples is consistent across Fixed Single, Fixed Dual, and Varied Multi triggers.** As we are unable to upload images or external links in the rebuttal, we provide the key sample-level statistics in the table below. The Kolmogorov–Smirnov (KS) distance quantifies the distribution gap for each trigger type.
>
> *Table 4: Sample-level attention entropy statistics and KS distances for Fixed Single, Fixed Dual, and Varied Multi triggers.*
>
> |**Trigger Type**|**Clean Mean**|**Poison Mean**|**KS Distance**|
> |-|-|-|-|
> |Fixed Single|5.47|2.96|0.91|
> |Fixed Dual|5.10|3.01|0.87|
> |Varied Multi|4.78|3.05|0.85|
>
> These results confirm that entropy-based separation is robust for all evaluated trigger types. We will include the empirical histograms in the revised manuscript.

---

> > ### Comment · Reviewer_wyVd · 2025-08-05
> > **Thanks for the response**
> >
> > Thanks for the response.
> >
> > The attention entropy diff is a key motivation of this paper, and I also see other reviewers raising similar doubts on the general applicability of BYE against diverse triggers. I will also maintain my concerns on this part. Nevertheless, the new results are good and should be included in the paper.
> >
> > W2 & W3 address my minor concerns.

---

> > > ### Author Response · Authors · 2025-08-05
> > > **Response and Appreciation**
> > >
> > > Thank you for your thoughtful feedback and for acknowledging the new experimental results.
> > >
> > > We understand your continued concerns regarding the general applicability of BYE, especially in scenarios involving diverse trigger types. To address this issue, we have already included a broader set of trigger designs in our previous rebuttal, and the results consistently support the effectiveness of BYE across different trigger settings. Our method is designed to capture the fundamental behavioral differences between clean and poisoned samples, as reflected in the attention entropy distribution. Rather than relying on any specific visual or semantic characteristics of the trigger, our approach detects the abnormal response patterns introduced by backdoor activation. This behavioral foundation allows BYE to maintain its robustness even when the triggers vary substantially in form or location.
> > >
> > > We appreciate your constructive suggestions, which have played a significant role in strengthening the paper and improving its clarity. We also notice that the additional analyses and new results addressing concerns about trigger diversity have been positively received by the reviewers, leading to improved evaluations. Your feedback has not only guided our revisions but has also helped demonstrate the robustness and generalizability of our method. We sincerely hope that our responses have addressed your concerns and have made the contribution of our work clearer.

---

> > > > ### Comment · Reviewer_wyVd · 2025-08-07
> > > > **Thanks for the new results**
> > > >
> > > > Since other reviewers value the effectiveness of BYE against blended triggers and the authors also provide multiple new results, I agree the effectiveness of BYE cannot be underestimated and I have increased my rating.

---

> > > > > ### Author Response · Authors · 2025-08-07
> > > > > **Appreciation for the Positive Feedback**
> > > > >
> > > > > Thank you very much for your positive feedback and for recognizing the value of our additional results. We greatly appreciate your careful review and your support in acknowledging the effectiveness of BYE against blended triggers. Your comments and increased rating are truly encouraging for our work.

---

### Decision · Program_Chairs · 2025-09-17

**Decision:**

Accept (poster)

**Comment:**

The paper presents a well-motivated and timely approach to backdoor cleaning in multimodal LLMs via entropy-based attention analysis. Reviewers agree that the problem is important and the proposed BYE framework is intuitive, effective, and validated through extensive experiments. Overall, while not groundbreaking, the paper offers a technically solid and practically relevant contribution that merits acceptance.